# Health inequalities in post-conflict settings: A systematic review

**Dieudonne Bwirire**[ID][1]*, **Rik Crutzen**[1], **Edmond Ntabe Namegabe**[2], **Rianne Letschert**[3], **Nanne de Vries**[1]

**1** Faculty of Health, Medicine and Life Science, Department of Health Promotion, CAPHRI Care and Public Health Research Institute, Maastricht University, Maastricht, The Netherlands, **2** Faculté de Santé et Développement Communautaires, Université Libre des Pays des Grands Lacs (ULPGL), Goma, Democratic Republic of the Congo, **3** Maastricht University, Maastricht, The Netherlands

* d.bwirire@maastrichtuniversity.nl

**Data Availability Statement:** All relevant data are within the manuscript and its Supporting information files.

**Funding:** The author(s) received no specific funding for this work.

## Abstract

Conflict can be a primary driver of health inequalities, but its impact on the distribution of social determinants of health is not very well documented. Also, there is limited evidence on the most suitable approaches aiming at addressing health inequalities in post-conflict settings. Thus, we undertook a systematic review of the literature concerning the current knowledge and knowledge gaps about structural determinants of health inequalities and assessed the effects of approaches aimed at addressing health inequalities in post-conflict settings. We performed a systematic search in bibliographic databases such as Web of Science, PubMed, and PsycINFO for relevant publications, as well as institutional websites that are relevant to this topic. The search was initiated in March 2018 and ultimately updated in December 2020. No time or geographical restrictions were applied. The quality of each study included in this review was independently assessed using criteria developed by CASP to assess all study types. *Sixty-two articles* were deemed eligible for analysis. The key findings were captured by the most vulnerable population groups, including the civilian population, women, children, internally displaced persons (IDPs), and people with symptoms of mental illness. A considerable range of approaches has been used to address health inequalities in post-conflict settings. These approaches include those used to address structural determinants of health inequalities which are accountable for the association between poverty, education, and health inequalities, the association between human rights and health inequalities, and the association between health inequalities and healthcare utilization patterns. However, these approaches may not be the most applicable in this environment. Given the multifactorial characteristics of health inequalities, it is important to work with the beneficiaries in developing a multi-sector approach and a strategy targeting long-term impacts by decision-makers at various levels. When addressing health inequalities in post-conflict settings, it may be best to combine approaches at different stages of the recovery process.

**Competing interests:** The authors have declared that no competing interests exist.

## Introduction

Promoting health and well-being for all is essential to sustainable development. In recent years, notable progress has been made in this regard, but significant challenges remain as health inequalities persist within and among countries [1]. For instance, it has been reported that in Low and Middle-Income Countries (LMIC), the coverage of reproductive and maternal health services is lower among socioeconomically disadvantaged sub-groups. The proportion of births attended by skilled health care personnel differed (by up to 80%) between the richest and poorest sub-groups within 83 study countries, and the use of modern contraception was at least twice as high among women with secondary schooling or higher than among women with no education in almost half of 71 study countries varying in national socio-economic status [2]. Conflict adversely impacts health and health systems, deteriorating structural and social determinants of health, including economic, social, cultural, environmental, and political systems [3].

Very few empirical studies have measured health equity in conflict-affected settings, but those have suggested that conflict has a significant impact [4]. Most importantly, there is limited evidence on appropriate approaches for health equity in such environments. To accomplish equity, the prerequisite is first to ascertain the presence of inequality so that targeted interventions can be planned, implemented, and monitored to determine the progress [5]. This systematic review aims to collate the current knowledge and identify knowledge gaps about structural determinants of health inequalities within post-conflict settings and assess the effects of approaches aimed at addressing structural determinants of health inequalities in this environment. It contributes to the body of knowledge on structural determinants of health inequalities in post-conflict settings which is also the main outcome of this study.

The social determinants of health in conflict settings reflect and further reinforce these inequalities and the vulnerability of those who are disadvantaged because of poverty, marginalization, and discrimination [6]. Three social determinants are specifically important in conflict settings: (i) the loss of human rights, which can be seen as the first and most important social determinant in a conflict situation; (ii) breaches of medical neutrality, in violation of the Geneva Convention, Article 18; and (iii) progression from stress to distress and disease that results from constant, unremitting exposure to a life-threatening situation [7]. Through its pervasive harmful effects on children, armed conflict is a negative social determinant of child health [8]. Nevertheless, the social determinants affecting children in crisis settings are the same as those affecting children in Low and Middle-Income Countries: (i) poor environment: lack of safe water and sanitation, low-quality housing; (ii) poor nutrition; and (iii) lack of access to health services [7].

There is evidence that conflict can be a primary driver of health inequalities [9]. Specifically, socio-economic status (as measured by education, income or occupation) is a key determinant of health. But, there are other more important determinants of health which are specific to the post-conflict environment [10], such as: 1) differential exposure—which may vary (by type, amount, and duration) between social groups 2) differential vulnerability—even when a given risk factor is distributed evenly across social groups, its impact on health may be unevenly distributed due to underlying differences between social groups in their vulnerability or susceptibility to that factor and 3) differential consequences—the impact of a certain health event differs depending on an individual's or family's socio-economic circumstances or health. Furthermore, life expectancy and mortality trends are familiar ways of measuring health status and monitoring health inequalities. Countries without basic data on mortality (including post-conflict countries), by socioeconomic indicators might have difficulties progressing their health equity agenda. The example of the DRC illustrates the links between conflict and regional inequalities

in mortality. It indicates that mortality rate was higher in unstable eastern provinces compared to stable provinces, and suggest that life expectancy was lower in more deprived areas.

Through a series of mortality surveys conducted between 2000 and 2004, the International Rescue Committee (IRC) has documented the humanitarian impact of war and conflict in the DRC. These studies estimated that 3.9 million people had died since 1998, arguably making the DRC's conflict the world's deadliest crisis since World War II. Less than 10 percent of all deaths were due to violence, with most deaths attributed to easily preventable and treatable health conditions (indirectly caused by conflict—and resulting in inequitable health) such as malaria, diarrhea, respiratory infections, and malnutrition [11]. Furthermore, deaths are often used as the primary indicator for the effect that conflict has on civilians; indeed, the number of these deaths declined in the DRC following an increase in security [12–14].

## Key concepts

To study health inequalities, it is of importance to clarify the following concepts: social determinants *versus* structural determinants of health; inequalities *versus* inequities in health; conflict *versus* post-conflict settings, effects of conflict on health equity, and right to health. Without clearly understanding those key concepts, no one can develop effective policies for health equity in this environment. This paper helps to define those key concepts that are needed to better understand health inequalities in post-conflict settings, such as:

### Social determinants versus Structural determinants of health

Because the concept of 'social determinants' refers simultaneously to both the determinants of health and the determinants of inequalities in health, we will expand our review of social determinants of health to include the terms "structural determinants of health" as we assess the literature on this topic. In this article, the term "structural determinants of health inequalities" refers specifically to the interplay between the socio-economic-political context, structural mechanisms generating social stratification, and the resulting socio-economic position of individuals. These mechanisms configure social groups' health opportunities based on their placement within hierarchies of power, prestige, and access to resources [15]. We hypothesize that structural determinants affect the distribution of resources and have the potential to influence health inequalities.

### Health inequalities versus health inequities

Health inequalities and health inequities have often been used interchangeably in academic and policy literature. For this paper, *health inequalities* refer to differences in the distribution of a specific factor (such as health status, income, opportunities) between different population groups, while *health inequities*, on the other hand, are inequalities in which the outcome is unnecessary and avoidable, as well as unjust and unfair [16]. Following clarification of their meanings and underlying values, we will now use both terms interchangeably throughout this paper.

### Conflict versus post-conflict settings

There is no consensus in the literature about the definition of post-conflict settings. From a global perspective, post-conflict has been defined as the aftermath of a situation that poses challenges severe enough to threaten the continued existence of an established political, economic, and social system. Post-conflict rarely means that violence and strife have ceased at a given moment in all corners of a country's territory [17]. For this systematic review, post-

conflict countries have been defined as having four characteristics: (i) the signing of a formal peace agreement; (ii) a process of political transition by-elections, military or civilian coups; (iii) increased levels of security; and (iv) a perception among national and international actors that there is an opportunity for peace and recovery [18]. Importantly, not all of these characteristics must be present simultaneously. Post-conflict countries can be volatile. Some countries, such as Angola, Liberia, and the Democratic Republic of Congo (DRC), repeatedly cycle between peace and war [18]. As regards the term 'conflict-affected': we use this term because, obviously, the interest in post-conflict areas arises from the very fact that the conflict-affected the health situation of the population.

## Right to health

Good health and well-being are considered human rights. Often, the right to health is associated with access to health care and the construction of health infrastructures. However, the right to health extends further; it should set the conditions for a right to health care systems, facilities, services, and goods (including the underlying determinants of health), which should be sufficiently available, physically and economically accessible, acceptable, and of good quality [19]. In recent years, increasing attention has been paid to good health and well-being by human rights treaty monitoring bodies included in various international instruments adopted under the auspices of the United Nations [20,21] or the World Health Organization [22]. Also, in the more recently acclaimed Sustainable Development Agenda, ambitions relating to health and well-being have been included, aiming to offer a new chance to ensure that everyone can access the highest health and health care standards, not just the wealthiest [1]. Nevertheless, failings and shortcomings affecting proper healthcare caused by, among others, armed conflict, widespread crime, persecution, corruption are daily realities. In contrast, several countries have emerged from war and made steady gains on the road to reconstruction, peacebuilding, and development, which led to reduced inequalities, vulnerability, and exclusion [22,23].

## Effects of conflict on health equity

Many studies have investigated the effects of conflict and its aftermath on health and health systems [24–28]. In general, it has been established that conflict harms health indicators [7,29]. However, the effect of conflict on health equity and the distribution of social determinants of health is not well documented. Also, the extent to which the most vulnerable population sub-groups experience health inequalities in post-conflict settings has not been systematically studied. Using existing literature, this paper addresses this gap.

# Materials and methods

This systematic review was conducted according to internationally accepted guidelines; we adhered to the Preferred Reporting Items for Systematic Reviews and Meta-Analysis statement (S1 Checklist. PRISMA) [30] and the PRISMA extension-checklist (PRISMA-E, which enabled an equity-focused systematic review) [31].

The outcome of interest was structural health inequalities and the population of interest consisted of the most vulnerable population groups [32,33] living in post-conflict areas such as the civilian population, women, children, IDPs, and people with symptoms of mental illness—as these are generally affected the most. The majority of the study population sub-groups was identified beforehand. In particular, we would not feel comfortable discussing the most vulnerable sub-groups in this environment without giving proper attention to women. The reasons for this are multifold: 1) traditional networks that form the foundation of women's social and family roles may be disrupted during the conflict, 2) women may also leave their communities

and husbands for other partners to guarantee a source of food for themselves and their families, and 3) women have to assume the role of the primary breadwinners and caretakers of their families—putting them more at risk [34,35]–for instance, the employment opportunities they could fall back on would be low-paying jobs because they may not meet the educational requirements for a better paying job [36].

While applying a stepwise data extraction approach, we found an additional sub-group of interest which consisted of the civilian population and added this afterward to the final sub-group selection. By civilian population, we mean people who are not active members of the security forces of the state, or members of an organized armed militia or opposition group. Government officials (e.g., members of parliament, governors, and councilors) are also excluded as they are instead seen as representatives of the government of a state. This sub-group is made up of individual civilians who do not belong to any of the various sub-groups that are typically recognized as being most vulnerable to inequalities in health.

## Search strategy

The search of the literature was initiated in March 2018 and ultimately updated in December 2020. The search strategy was comprehensive to allow for a wide variety of study designs and interpretations of structural determinants of health inequalities in post-conflict environments to be included.

We used the PROGRESS-Plus as an organizing framework to assess the structural determinants of health inequalities in post-conflict settings. Using PROGRESS-Plus enables the inclusion of a range of factors and circumvents reliance on any single measure to assess inequities, as this is insufficient. The PROGRESS-Plus acronym stands for: Place of residence, Race/ethnicity/culture/language, Occupation, Gender/Sex, Religion, Education, Socio-economic status, and Social capital [37], the factors identified by WHO as relevant stratifiers that can be used to define social groups [38].

We searched the following bibliographic databases: Web of Science (S3 File), PubMed, and PsycINFO for relevant publications, as well as institutional websites that are relevant to this topic, including those of the World Health Organisation, the Health and Fragile States Network, Evidence aid and the 3i.e (International Initiative for Impact Evaluations) database of systematic reviews. References from included studies were checked to identify other relevant studies. Within the bibliographic databases, we searched for published and unpublished articles and abstracts using three sets of key concepts i.e., post-war, inequality, and health [39], in combination with any of the above-mentioned PROGRESS-Plus factors. Depending on each database's specifications and to increase the sensitivity to retrieve information, MeSH (Medical Subject Headings) and non-MeSH keywords were used as appropriate. No time or geographical restrictions were applied.

The grey literature was collected from various relevant conferences on this topic (e.g., the Post-Conflict Transitions symposium organized by the World Bank). Lastly, we screened relevant public health journals (e.g., Bulletin of the World Health Organisation; PubMed indexed public health journals, etc.). Given a large number of potentially relevant grey literature on the topic, only peer-reviewed reports were included. We finally contacted experts for advice (e.g., the plan to use PRISMA standard in combination with PRISMA-E was discussed with the editor in chief of the Campbell Collaboration Library).

## Study selection

The first author and an information specialist from Maastricht University independently screened the titles and abstracts of all references retrieved by the search strategy to exclude

irrelevant results. Potentially relevant studies about structural determinants of health inequalities in post-conflict settings were primarily assessed and reviewed for eligibility based on specified study characteristics. Publications had to meet the following initial criteria: a) post-conflict environment was explicitly described as study setting; b) the most vulnerable sub-groups were considered as study participants, c) assessing and addressing the structural determinants of health inequalities were seen as interventions, d) qualitative or quantitative (or mixed methods) studies were used. To be included in the final review, publications had to meet the following eligibility criteria (S1 File): (i) contain at least one of the three key concepts (post-war, inequality, and health) (S2 File); (ii) include at least one PROGRESS-Plus acronym of interest and, (iii) have all of the above-described study characteristics (a to d). Studies about structural determinants of health inequalities in contexts other than post-conflict environments were excluded (e.g., natural disasters/post-tsunami, terror victims, intimate partner violence). Abstract-only studies, conference abstracts, studies about ongoing conflict settings, and studies on serving or former military personnel were excluded. All other reasons for exclusion have been documented in a PRISMA chart, and available in supporting information (S1 Fig).

The PRISMA flow diagram represents the search strategy and the study selection process. From the initial search in different bibliographic databases, a total of 4,742 potentially relevant studies were identified; 1,743 studies were duplicates and therefore removed. For the remaining 2,999 studies, 1,773 were excluded based on title screening. Of the remaining 1,226 studies, 886 studies were excluded based on abstracts.

After removing duplicates and screening the titles and abstracts, 340 full-text papers were retrieved and read. Of these, 284 studies were excluded, based on diverse reasons specified in the PRISMA flow diagram [S1 Fig], and 55 articles were selected and included in the final review. The above searches were supplemented with a systematic search of the International Initiative for Impact Evaluations (3i.e) database of systematic reviews. We identified 107 potentially relevant studies from which 7 articles were included in the final review.

As shown in S1 Fig, 62 publications about structural determinants of health inequalities in post-conflict settings were identified and included in this review. All retrieved articles were checked for inclusion and exclusion criteria by two reviewers (DB, GF).

## Data extraction and analysis

All identified indexed records concerning this topic were downloaded into EndNote software by the first author and cross-checked by a second reviewer. Discrepancies and disagreements were discussed. A third reviewer mediated when consensus could not be reached.

A stepwise approach for data extraction was adopted as follows: i) filtering by the title and abstracts for relevance, followed by ii) full-text retrieval and review of all potentially relevant publications by a single reviewer (DB) to determine their eligibility for inclusion in this review iii) reference lists of included articles were manually screened to identify additional studies that our strategy could have missed. We also applied the same approach to the grey literature (S1 Fig).

Based on the study objectives, a predesigned data extraction sheet (S1 Table) was used for retrieving information from the selected studies, including author details, topic, study setting, and population, the phenomenon of interest, study design, type of health inequalities studied, evaluation (study outcomes), research type, study results, and study quality. Segments of text describing approaches/interventions to reduce health inequality within the most vulnerable sub-groups were also extracted.

We analyzed relevant information from included papers and synthesized this using different steps, including i) the textual description of the study, which is the process of identification

of key findings from each included primary study; ii) the grouping and clustering by organizing the key findings from individual studies into themes and iii) the content analysis for translating data was thoughtfully applied resulting in the generation of analytical themes.

Data was described, analyzed, criticized, summarized by the most vulnerable sub-groups, and reported as part of the evidence in this systematic review.

Due to the heterogeneity of study settings, populations, and interventions (aiming to reduce health inequalities in post-conflict areas) captured in the included publications, it was impossible to conduct a meta-analysis.

## Quality assessment

The quality of the included studies was assessed using the adapted critical appraisal skills program (CASP) checklist developed for systematic reviews [40], consisting of 10 questions that can be answered with "Yes," "No," and "Can't tell" about the methodological and reporting issues. This original checklist was designed to be used as an educational pedagogic tool, thus does not suggest a scoring system. Therefore, included studies were assigned an overall score of 'high', 'medium' or 'low' based on methodological quality rating of included papers- a summary is presented in S4 Table. Additionally, we assessed the quality of the studies using different areas such as the research question, study design, data collection methods, data analysis, study findings, and ethics [41,42]. Quality was not used as an exclusion criterion.

## Results and discussion

All included articles were published in English. Most of these were published from 2002 onwards, with at least more than one article published each year. On average, 5 articles were published each year in 2008, 2012, 2014, and 2019 (S2 Table). There were peaks in publications in 2013 (10) and 2015 (9).

Among the included publications, 24 were quantitative studies, 22 were qualitative, and 16 were mixed methods design. Of the total of 62 publications included in our final review, 57 studies were conducted in post-conflict settings. There were 6 studies in which the setting was not always clearly defined.

The majority of the studies were based on data collected in sub-Saharan Africa (27/62 total publications; 43%), followed by South Asia (7/62 total publications; 11%)–suggesting that most conflicts have occurred in both geographic regions in the past decade, and the Middle East (6/62 total publications; 9%), and North Africa region (6/62 total publications; 9%). Small numbers of papers emanated from three regions, including Asia Pacific, South America, and Europe—an indicator that recent peace agreements have been reached in these regions. Only one study collected data in the Russia and Eurasia region.

Publications identified by our search varied with regards to population, context, design, outcome measures, or whether the primary study focus was on health, inequality, post-war, or any PROGRESS-Plus factor of interest, making comparisons difficult. To capture this significant heterogeneity, a narrative synthesis approach was adopted.

The methodological quality of each study included in this review was independently assessed using customized criteria developed by CASP to assess all study types. The quality of the studies ranged from high-quality [43–58], medium-quality [59–95], and low-quality [96–103].

Key findings regarding current knowledge and knowledge gaps about structural determinants of health inequalities in post-conflict settings are presented by the most vulnerable population sub-groups, including (i) the civilian population, (ii) women, (iii) children, (iv) IDPs,

and (v) the people with symptoms of mental illness. Additionally, key approaches aimed at addressing health inequalities in this setting are presented.

## Structural determinants of health inequalities by the most vulnerable population sub-groups:

**Civilian population.** The social determinants of health such as infrastructure, poverty, housing, education, and discrimination affect the context in which people are born, grow, live, work, and age and their influence on health [58]; they are major contributors to inequalities in health [104]. We found thirty studies [43,45–47,49,51,55–60,63,66,67,70,71,74,76,80,83,86,89,90,93,94,97,100,101,103] examining the multiple interrelated and complex social determinants of health in conflict and post-conflict environments. Most of these studies [45,46,51,56–58,67,89,93,97,100,103] report an association between poverty, education, and health inequality. For instance, Bisogno and Chong [45] identified that the higher the level of education attained, the higher the earnings.

Within the healthcare setting, education may also be linked to self-efficacy. For example, discussing care options with the healthcare provider and insisting on receiving adequate care [89], suggesting that the higher the level of patient's education, the greater confidence will be expressed, a finding that is consistent with the literature on the subject [105]. Similarly, a cross-sectional survey examining the factors contributing to the resilience of women in the aftermath of Peruvian demonstrated that low levels of education were all associated with lower resilience [88].

Twelve studies [47,49,55,63,80,83,86,90,93,100,101,103] report on healthcare utilization patterns and how a well-functioning and equitable health system can promote health equity in post-conflict. In particular, one cross-sectional study examining health care utilization patterns for mental, behavioral, and emotional problems in Georgia [47], reported that only just over a third of those with a current mental disorder sought any assistance from health services. Despite meeting the criteria for a current mental health disorder, the other persons did not use health services quite simply because they did not report the presence of mental health problems. This suggests that the demand for facility-based services may decrease in environments where there has been political instability regardless of their availability; thus rebuilding trust in the health system may be as important as rebuilding facilities and improving which services are available [90].

Two studies [49,76] describe the presence of an accountability mechanism and community participation as significant program facilitators. The first study assessed community scorecards feasibility in a post-conflict context through the joint engagement of service providers and community members in the design of patient-centered services. It concluded that joint interface meetings facilitated transparent dialogue between the community and service providers resulting in creative and participatory problem-solving mechanisms and mobilization of resources. The second study examined the accountability mechanisms in a community-driven reconstruction program and found that accountability was translated into informal and oral, rather than formal and written, mechanisms. The findings suggest that an adaptation to the local context enhances the working of accountability but restricts its possible impact as it becomes part of local power dynamics in the process.

Need is the primary determinant of healthcare utilization. In post-conflict settings, health services should be available to users in proportion to need rather than social status, and financing for health services should require proportionately fewer contributions from those with the lowest incomes [53]. Furthermore, healthcare utilization is determined by the need for care, by whether people know that they need care, by whether they trust and want to obtain care, and

by whether care can be accessed. Other factors (e.g., poverty, geographic area of residence, sex, age, and language) are also important [106]. More evidence on the instrumental role that health systems can play in reducing inequities is available from countries at peace where integrated health systems approaches have been shown to reduce disparities in access to care and improve financial protection from catastrophic health care costs for the poor [107–110].

**Women.** It is widely recognized that armed conflict has a particularly negative impact on women's lives [111,112]. They suffer disproportionate effects during and after the war, such as inaccessibility of health and education services and sexual violence [113]. As such, "Gender affects exposure to situations which have an impact on health, dictates who has access to health care services and influences its planning and provision [114]—Pg.2". We found ten studies [52,61,68,69,73,84,88,92,95,99] explicitly reporting on women. Ultimately, there is some available evidence on the links between social determinants of women's health and health inequalities in post-conflict settings. For example, Hynes [99] point out how women are harmed by war-related disintegration of health, education, and social services and by the loss of essential environmental assets, such as potable water, sanitation, land, and food.

A study investigating whether differences in women's rights or gender inequalities were associated with stroke mortality at the country-level confirmed a consistent association between women's rights inequalities and worse health outcomes [73]. The result indicates that gender inequality status is associated with women's stroke outcomes.

Two studies [61,68] examined gender justice in post-conflict areas. The first study shows that underlying power inequalities and gender ideologies continue to act against changes in gender relations. The findings suggest a link between men's rigid adherence to gender roles and expectations and the incidence of sexual and gender-based violence. The other study suggests if gender justice is to be improved, it is necessary to place women, their experiences, and their opinions at the center of thinking about transitional justice mechanisms.

Another study assessing inequalities in access to reproductive and maternal health services in Colombia shows that failure to provide these services to females undermines the goal of universal health coverage [84]. This finding suggests a pattern of marginal exclusion and cumulative patterns of inequality in the reproductive and maternal health care service provided to females affected by armed conflict. Elsewhere in Timor-Leste, factors associated with the non-utilization of health services for childbirth in post-conflict were investigated [52]. The authors found that women residing in rural areas were almost three times more likely not to utilize a health facility for childbirth than their urban counterparts. The authors cite as a possible explanation that women living in urban areas have better transportation, economic status, and access to hospital and health services, which enhance their opportunity for facility delivery.

Although a substantial amount of literature has been published on the provision of Sexual and Reproductive Health (SRH) services in post-conflict settings, there is a lot we do not know yet about how to address the problem. To illustrate, a recent systematic review of monitoring and evaluation indicators for SRH demonstrates that rigorous reporting on a core list of indicators is a prerequisite for identifying the changes that a country wants to see concerning SRH and the investments required to save lives [115].

Women and men have different exposures to situations that affect their health and access to health care, they also have differential power to influence decisions regarding the provision of health services. Moreover, women and men have different needs, including unequal access to and control over power, resources, human rights, institutions, and the justice system. According to Sorenson [116], health care and other social facilities remain inadequate for women's health in the post-conflict period. As such, they face new challenges (e.g. their traditional roles and relationship) and inherit additional responsibilities. The social transformation occurring in the post-conflict context opens up opportunities that should not be missed for women.

Significant differences in sociodemographic and lifestyle characteristics between men and women were documented in northern Uganda in a cross-sectional community survey. The authors found that more women were divorced and widowed than men; underweight was higher among men than women, with elderly and young men being more likely to be underweight [86]. Further studies [117,118] report the same trends, with underweight being more prevalent among men and overweight more prevalent among women. These findings highlight the importance of continuous health and nutritional assessments of all population groups that reflect local social determinants and family structures.

**Children.** Seven studies [48,62,64,78,95,96,102] explored current knowledge and knowledge gaps of social determinants of health and health inequality in children in post-conflict zones. Two studies [48,62] show how children's well-being was impacted by the scarcity of resources. This included the shortage of electricity, shortage of clean drinking water, disturbed community life, and unsafe streets and playgrounds in the first study. In the second study, the scarcity of resources was characterized by the chronic lack of specialized facilities to address the complex issue of severe malnutrition.

Two other studies [78,96] describe the post-war educational system. The first study discusses post-war education and gender equity in Sierra Leone [78], it shows that post-war educational reconstruction is unlikely to rectify gender inequities that remain well established within mainstream schooling and in the broader social context. The authors argue that the capacity of education to contribute to gender-based change has not been entirely muted. The second study examines the link between schooling, conflict, and gender differentials in these links [96]. The results indicate that boys' schooling was more negatively affected than that of girls because boys are withdrawn by their families to contribute to the household income or because they are drafted into (armed) serving in the conflict.

In a study of the effects of the conflict on mortality both in utero and during the first year of life in the DRC, Dagnelie and colleagues [64] show that in utero exposure to conflict negatively affects the number of males (by a reduction of the number of male fetuses) reaching birth. The authors show that conflict exposure increases infant mortality specifically among girls. Possible explanations for such an outcome are in utero biological factors. In another study discussing why achieving gender equality is of fundamental importance to improve the health and well-being of future generations, Roseboom [95] notes that prenatal exposure to violence has been associated with long-term effects on children's health, especially on brain structure and neural function, that can still be detected many years later. A case study examining the role that young boys and girls assume in negotiating household poverty and enhancing their livelihood opportunities in small-scale mining communities from rural sub-Saharan Africa [102] reveals that the implications of their involvement are often complex than international children's rights advocates understand them to be suggesting that Western notions of "progress" and development often do not match up with the complex realities or competing visions of local people.

Little is known about how the circumstances of a post-conflict setting could affect the health status of children. A recent systematic review of the effects of armed conflict on child health and development [8] shows some gaps in the medium and long-term effects. The authors report limited data on child health and further supports the need for research on how conflict and structural inequalities affect children and subsequent generations. According to the Human Security Centre, child mortality is high in post-conflict countries due to food insecurity leading to malnutrition and the disruption of the water supply and sanitation [119]. This suggests that children are more vulnerable during and after conflicts, as a consequence of a quick transition from conflict to post-conflict in these environments.

**Internally displaced persons (IDPs).** Most of the time, IDPs are unable to meet their basic needs. They have specific burdens over belonging, housing, occupation, welfare, security, and loss of communities.

Eight studies [44,65,72,75,81,82,85,120] involving IDPs were identified, they collate some evidence of knowledge and gaps about social determinants of health of persons who have been forced to leave their homes due to armed conflict but remained within the borders of their countries. The majority of these studies [65,72,75,81,82] have reported a link between IDPs and poor health outcomes. Specifically, three studies [65,72,81] have reported a long-term effect of conflict on IDP's health and health care provision. The first study in Bosnia-Herzegovina claims that IDPs remain a forgotten and vulnerable sub-group long after the conflict has ceased and illustrates how the provision and utilization of healthcare services can alleviate some of this vulnerability [72]. The second study investigated the long-term health impacts of internal displacement [65], this study indicated that the IDPs are more likely to report poor self-rated health (SRH) than non-IDPs, suggesting that internally displaced experience poorer health than non-displaced. The third study illustrated medical care among IDPs, in South Sudan [81]. The findings identified that the IDPs have been attracted to services in the hospital because it was representative of improved access to healthcare that could control the health effects associated with displacement and war.

Two studies involving IDPs [44,85] have investigated the relationship between poverty and health inequality in Uganda. The first study explored the relationship between conflict, education, and the intergenerational transmission of poverty. The results confirm that conflict has long-term and intergenerational impacts on well-being and livelihoods [44]. It further confirms that education can limit the poverty impact of conflict on households and support a speedier post-conflict recovery. The other study examined factors that contributed to early relationships and informal marriages in post-conflict settings [85]. The findings note that fundamental shifts in economies and family relationships combined with structural changes encountered in settlements resulted in changed relationships and marriage patterns. In this study, poverty had affected the views and perceptions of the participants around relationships and marriage.

One cross-sectional study of the prevalence of mental disorders in post-conflict primary care attendees (with the majority of participants reporting more than one displacement experience) in the Northern Province of Sri Lanka reports unmet mental health needs in the region and the necessity to address these [120], suggesting that healthcare provision for IDPs remains inadequate and underfunded. This is consistent with previous research findings in northern Uganda where extremely high mortality rates and physical and mental illness amongst IDPs have been recorded [121].

**People with symptoms of mental illness.** We found seven studies [50,54,55,77,79,87,91] examining social determinants of mental health inequalities in post-conflict settings.

Two studies [77,79] assessed the prevalence of mental disorders in post-conflict environments. The first reported a cross-sectional epidemiological study in Nepal where high prevalence rates of mental disorders were found, especially among women, the elderly, widows/widowers, or people separated from their partners. Specifically, depression and anxiety prevalence rates were high compared to other epidemiological studies in high-income settings and comparable to the prevalence rates found in other conflict-affected settings. Study findings underscore the need to address the current lack of mental health care resources in post-conflict rural Nepal, especially for the most vulnerable populations. The second study examined the prevalence and factors associated with Posttraumatic Stress Disorder (PTSD) in northern Uganda. This study found that the prevalence rate of PTSD in the study communities was unacceptably high.

Elsewhere, a study analyzed the relation between exposure to the armed conflict and violence with mental health disorders and assessed the extent and determinants of socioeconomic inequalities in mental health-related to differential exposure to the conflict and violence in Colombia [55]. The findings indicate that victims of armed conflict have a higher probability of suffering mental health disorders compared to non-victims. It shows that differential exposure to armed conflict among lower socio-economic sub-groups explains high proportions of total inequality in mental health disorders.

Two studies [54,87] elaborate on resilience in a post-war context. The first study [54] evaluates a school-based primary prevention intervention designed to promote adolescents' coping in the immediate aftermath of war exposure. Study findings provide validity for the effectiveness of the intervention in promoting its two targeted resilience factors of mobilization of support and self-efficacy. These findings indicate the need to apply interventions to children at the earliest possible time post-exposure. The second study [87] reports a qualitative, ecological, psychosocial ethnography study in Northern Sri Lanka. It explores psychosocial problems faced by families and communities and the associated risk and protective factors so that practical and effective community-based interventions can be recommended to rebuild strengths, coping strategies, and resilience. Study findings indicate that complex mental health and psychosocial problems at the individual, family, and community levels in a post-conflict context were found to impair recovery.

One qualitative study [50] explored concepts related to mental distress and coping strategies among adults in Burundi. The findings put forward that a general lack of knowledge and information on basic mental health concepts hinders treatment opportunities. For example, most individuals with symptoms of mental illness tend to stay home without receiving any treatment; however, some people seek help in the available infrastructure within their communities [50]. This suggests that more research is needed to know the characteristics of these available infrastructures and how a public mental health model could build from them.

One study exploring the mental health and psychosocial problems of displaced persons living in refugee settings reveals that one consequence of living in situations of pervasive adversity caused by experiences of discrimination, inequity, and violence is poor mental health and psychosocial well-being [91].

Both short- and long-term negative consequences of exposure to the conflict have been documented. Short-term effects included distress, shock, fear, phobic avoidance of public places, anger and emotional pain [122,123], and aggressive behavior [124], while long-term effects included PTSD [125,126], anxiety, and depression [127–129], and sub-clinical symptoms [130]. These findings suggest the need to apply interventions at the earliest possible time post-exposure to traumatic events associated with war and its aftermath affects mental health at both the individual and community levels [54].

## Key approaches aimed at addressing health inequalities in post-conflict settings

We found seventeen publications [46,49,53–56,58,70,80,82,84,89,91,93,97,100,120] assessing approaches aimed to address social determinants of health inequalities in post-conflict settings. The majority of these publications set forth approaches that are believed to be accountable for the association between poverty, education, and health inequalities, such as (i) multi-dimensional local interventions to break the cycles of poverty [46]; (ii) the 'diseases of the poor' versus 'diseases of the rich' criteria used to prioritize which services to cover [93]; (iii) multi-sectoral approaches [56,58] that pay attention to the political, social and economic determinants of health; and (iv) combined approaches [89,97,100] with different systems being more

appropriate at different stages of post-conflict recovery. Notably, two studies [70,100] provide evidence that incorporating economic components into public health interventions is essential for effectiveness across disciplines of global health. Other publications sum up approaches used to address social determinants of health inequalities in relation to healthcare utilization patterns. For instance, (i) interventions that increased quality and access of mental health treatments among victims of the conflict, also improved mental health among victims and significantly reduced inequalities in mental health [55]; (ii) the Community Score Cards (CSC)—a valuable tool for enhancing social accountability for patient-centered care facilitated the flow of information between citizens and service providers [49]; and (iii) a Community Health Workers program increased access to the primary health care services [80].

Other studies outline approaches which are accountable for the association between human rights and health inequalities, such as (i) good governance and respect for human rights reinforce each and influence people's health [82]; (ii) the Ministry of Health acting as a steward of the health system, by contracting with NGOs to provide services but remaining responsible for the formulation of health policies (e.g., during the development of the Basic Package of Health Services) [53]; (iii) 'for the people' drive Community-Based Rehabilitation model (CBR), with equality of rights for people with disabilities (PWD) and their integration into the mainstream of community life [97], and (iv) training of primary care practitioners using a scaled-up intervention based on the WHO Mental Health Gap Action Programme (mhGAP) to address unmet mental health needs [120].

While a range of approaches aimed at addressing health inequalities in stable countries may not be replicable in post-conflict settings, some approaches appear to be promising in a post-conflict environment. For instance, education is essential for instilling stability and normalcy in the lives of children who have experienced war [131]. We found that multi-sectoral approaches and approaches incorporating economic and public health interventions and services prioritization criteria have been successfully used to address this association between poverty, education, and health inequalities.

When women have a role in the decision-making over household economics, the children in the household are healthier and more likely to attend school [132]. Thus, an important opportunity exists to use the microfinance program as a mechanism to provide information to household members on women's safety and health topics through research [70].

We know now that enhancing community participation can promote positive health practices and motivate the most vulnerable citizens to access health services, better advocate for their needs, and reinforce accountability [133,134]. In Afghanistan, the Community Health Workers (CHW) program has increased access to primary health care services [80]. This finding is consistent with what has been shown in other countries (such as Ethiopia, Iran, Pakistan, and Brazil) with national CHW programs [135–138]. We further found approaches aiming to address the association between human rights and health inequalities. For instance, the Basic Package of Health Services (BPHS) has become a model for the reconstruction of health systems in post-conflict countries. It is a government's commitment to the population to provide broad coverage of essential services in Afghanistan [139]. This finding suggests that implementation of the BPHS program has resulted in increased access to basic health services for the entire population, especially those in rural areas, thus suggesting improved equity of care [4,140].

Community Driven Reconstruction (CDR) programs have been particularly popular in post-conflict areas. Advocates argue that the model is strong and effective. According to the World Bank, "CDR operations produce two types of results: more and better-distributed assets, and stronger, more responsive institutions—Page 18". The results from a study of the effects of a large-scale CDR program on the economic well-being and the socio-political

attitudes and behaviors of the populations in the DRC were negative [71]. These findings present a challenge to the CDR model and its ability to produce the social and economic impacts that advocates attribute to it [66].

## Study strengths and limitations

Although many studies on health inequalities have been conducted in *stable* contexts, this review is, as far as we know, the first about structural determinants of health inequalities in post-conflict settings. It collates the current knowledge and knowledge gaps and assesses the effects of approaches aimed at addressing structural determinants of health inequalities. Therefore, it adds to a relatively small body of existing literature investigating this topic in a *post-conflict* setting. To identify relevant publications, we conducted a comprehensive and systematic review of the literature. We included quantitative, qualitative, mixed methods and other types of studies in the final review. Of particular interest was the inclusion of studies using at least one PROGRESS-plus factor in the final selection, making this an equity-driven process. There are, however, several limitations to consider when interpreting the findings of the current review.

First, we sought to ascertain the presence of health inequalities so that targeted interventions could be planned, implemented, and monitored. By doing so, we found significant heterogeneity of settings, populations, interventions, and outcomes of interest in the studies that were reviewed, making international comparisons of interventions and outcomes difficult. As such, we used the narrative synthesis to bring together the findings from all included studies; however, it should be noted that this approach is not underpinned by an authoritative body of knowledge or on reliable and rigorous techniques developed and tested over time. In the absence of such a body of knowledge, there is a possibility that systematic reviews adopting a narrative approach to synthesis might be prone to bias. This must be weighed against the counter-argument that we selected our evidence judiciously and purposively with an eye to what is relevant for our main review questions, including any future research questions we would like to address. Instead, the quality of the included studies was assessed systematically.

Second, as with any systematic review, there were concerns about whether we might have missed certain evidence. To be as conclusive as possible, a significant effort was made by applying a rigorous search strategy and identifying articles from scientific and professional databases only.

Third, although we agree and recognize LGBTQ+ (Lesbian, Gay, Bisexual, Trans, Queer, and others) people as belonging to the most vulnerable sub-groups in post-conflict settings—this sub-group was not included in our systematic review because LGBTQ+ issues require special attention. For the same reason, they could not *reasonably* be easily summarized in the current manuscript. To successfully tackle LGBTQ+ issues, special consideration must be given to the particular needs and risks faced by individuals based on their sexual orientation and gender identity. Future systematic reviews on health inequalities within the post-conflict environment should also include LGBTQ+ people as a vulnerable sub-group.

Despite the limitations noted above, some clear themes about structural health inequalities in post-conflict settings were evident. Ultimately, the breadth of the papers and grey literature reviewed, the study selection methods, and the appraisal process altogether provided a substantial body of evidence that supports our key findings reported in the review.

## Conclusion and future directions

This systematic review documents the existing evidence and knowledge gaps about structural health inequalities in post-conflict settings. The results of our review assert the urgent need for

addressing identified knowledge gaps as well as the need for developing appropriate approaches aiming at addressing health inequalities in this environment. To shape adequate responses, steps forward should include ways and means to address health inequalities successfully. In particular, priority should be given to studies from the "post-conflict real world". Such studies should aim to: (1) ensure that the voices of the most vulnerable groups are heard, their active participation as agents of change is guaranteed, and local norms are respected. (2) exploring the novel design of a health system that would ensure both geographic and social equity.

A considerable range of approaches has been used to address health inequalities in post-conflict settings, including those used to address structural determinants that are accountable for the association between poverty, education, and health inequalities, the association between human rights and health inequalities, and the structural determinants of health inequalities in relation to healthcare utilization patterns. However, these approaches may not be the most suitable in this environment. They may not have involved vulnerable groups in decision-making processes and the implementation of strategies targeting relevant drivers of health inequalities.

Given the multifactorial characteristics of health inequalities, it is crucial to work with the beneficiaries in developing a multi-sector approach and a strategy targeting long-term impacts by decision-makers at various levels. Additionally, it may be best to combine approaches aimed to address health inequalities at different stages of the post-conflict recovery process.

This review has strengthened our conviction that there is room for innovation in developing potentially effective "real world" approaches, targetting and involving the most vulnerable groups in addressing health inequalities.

## Supporting information

**S1 Checklist. PRISMA 2009.**
(DOC)

**S1 Fig. PRISMA flow diagram of selection process.**
(DOC)

**S1 Table. Data extraction sheet information from selected studies.**
(XLSX)

**S2 Table. Characteristics of studies included.**
(XLSX)

**S3 Table. Approaches aimed at addressing health inequalities.**
(XLSX)

**S4 Table. Individual quality assessment for each study included.**
(XLSX)

**S1 File. Eligibility criteria.**
(DOCX)

**S2 File. Key concepts and search terms.**
(DOCX)

**S3 File. Web of Science search strategy.**
(DOCX)

## Acknowledgments

The authors gratefully acknowledge the following colleagues for their assistance in the study implementation and conduct:

*Gregor Franssen*—the Information Specialist University of Maastricht for support during bibliographic database searches.

*Stefan Jongen*—the Information Specialist University of Maastricht for EndNote support.

*Suat Tuzgöl*—Researchsoftware for support on the correct use of EndNote software thoroughly the process.

## Author Contributions

**Conceptualization:** Dieudonne Bwirire.

**Data curation:** Dieudonne Bwirire.

**Formal analysis:** Dieudonne Bwirire.

**Investigation:** Dieudonne Bwirire.

**Methodology:** Dieudonne Bwirire.

**Project administration:** Dieudonne Bwirire.

**Supervision:** Rik Crutzen.

**Writing – original draft:** Dieudonne Bwirire.

**Writing – review & editing:** Rik Crutzen, Edmond Ntabe Namegabe, Rianne Letschert, Nanne de Vries.

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
