## [Decision Letter · Decision Letter 0]

27 Sep 2021

PONE-D-21-08382

Health inequalities in post-conflict settings: A systematic review.

PLOS ONE

Dear Dr. Bwirire,

Thank you for submitting your manuscript to PLOS ONE. After careful consideration, we feel that it has merit but does not fully meet PLOS ONE’s publication criteria as it currently stands. Therefore, we invite you to submit a revised version of the manuscript that addresses the points raised during the review process.

We look forward to receiving your revised manuscript.

Kind regards,

Jakob Pietschnig, PhD, MSc

Academic Editor

PLOS ONE

Journal Requirements:

3. Please confirm that you have included all items recommended in the PRISMA checklist including: 

- a Supplemental file of the results of the individual components of the quality assessment, not just the overall score, for each study included. 

- details of reasons for study exclusions in the PRISMA flowchart and number of studies excluded for each reason.

- See https://journals.plos.org/plosmedicine/article?id=10.1371/journal.pmed.1000100#pmed-1000100-t003 for guidance on reporting. 

Thank you.

5. Please remove all personal information, ensure that the data shared are in accordance with participant consent, and re-upload a fully anonymized data set. 

6. Thank you for submitting the above manuscript to PLOS ONE. During our internal evaluation of the manuscript, we found significant text overlap between your submission and the following previously published works, some of which you are an author.

https://www.scribd.com/document/328300724/Estadisticas-de-Salud-Mundial

https://applications.emro.who.int/dsaf/dsa955.pdf

https://journals.plos.org/plosone/article?id=10.1371%2Fjournal.pone.0210071

https://www.who.int/social_determinants/links/events/conflicts_and_sdh_emro_revison_06_2007.pdf

https://apps.who.int/disasters/repo/8678.pdf

https://www.pure.ed.ac.uk/ws/files/19589731/Palmer_J._2014_._Changing_landscapes_changing_practice.pdf

https://www.coe.int/en/web/genderequality/what-is-gender-mainstreaming

https://onlinelibrary.wiley.com/doi/pdf/10.1111/tmi.12708

https://www.raiseinitiative.org/library/download.php?id=495

https://www.tandfonline.com/doi/abs/10.1080/17441692.2013.819587?journalCode=rgph20

Please revise the manuscript to rephrase the duplicated text, cite your sources, and provide details as to how the current manuscript advances on previous work. Please note that further consideration is dependent on the submission of a manuscript that addresses these concerns about the overlap in text with published work.

Reviewers' comments:

Reviewer's Responses to Questions

**Comments to the Author**

1. Is the manuscript technically sound, and do the data support the conclusions?

Reviewer #1: Yes

Reviewer #2: Yes

Reviewer #3: Yes

2. Has the statistical analysis been performed appropriately and rigorously? 

Reviewer #1: Yes

Reviewer #2: N/A

Reviewer #3: Yes

3. Have the authors made all data underlying the findings in their manuscript fully available?

Reviewer #1: Yes

Reviewer #2: Yes

Reviewer #3: Yes

4. Is the manuscript presented in an intelligible fashion and written in standard English?

Reviewer #1: Yes

Reviewer #2: Yes

Reviewer #3: Yes

5. Review Comments to the Author

Reviewer #1: Dear Editors,

Thank you for the opportunity to review this systematic review of research on health inequalities in post-conflict settings.

The article has many merits as it stands now. Relevant concepts (such as health inequalities, post-conflict, PROGRESS-Plus) are clearly defined. The aims of the study are appropriate, relevant, and narrow enough for the systematic review. My impression is that the authors conducted a rigorous, unbiased and comprehensive search for all studies according to the search strategy, which was clearly described and seems suitable for the aims.

Given the good shape of the article in its current condition, I believe the authors are capable of submitting an improved revised manuscript.

Some minor but important improvements are suggested below.

1) On page 6, the authors write “… the population of interest consisted of the most vulnerable population groups [34, 35] living in post-conflict areas such as women, children, internally displaced persons (IDPs), and the mentally ill – as these are generally affected the most.” Why were these groups identified as most vulnerable? In the results section it appears as if this choice was data-driven (e.g. “It is widely recognized that armed conflict has a particularly negative impact on women's lives [105, 106]” on page 13 and the paragraph starting with “Women and men have different needs, living conditions, and circumstances, including unequal access to and control over power, resources, human rights, and institutions, including the justice system” on page 21). But the earlier statement on page 6 makes it sound like it is theory-driven. Some clarification as to what guided the sub-group selection would be helpful.

I agree children, IPDs and the mentally ill are likely particularly vulnerable, but why are women included? Since women may make up far more than 50 percent of the population in post-conflict settings due to high male battle-related mortality, what is the logic behind identifying them as one of the most vulnerable groups? Feminist security scholars such as Cynthia Enloe have criticized the merged category “womenandchildren” for removing women’s agency by only portraying them as mindless victims. While I do not necessarily disagree that women are a vulnerable group, a clearer explanation to this effect would help.

A specific focus on women also risks sidelining men from post-conflict interventions, as argued on page 21. The study from Uganda [86] suggests that women were more likely to be widowed, which also likely means that men are more likely to die in war. Is that not also a vulnerability? Additionally, past research has shown that if post-war aid only focuses on women, it may drive intimate partner violence perpetrated by men who have lost their socioeconomic position as main breadwinners. See Sengupta, Anasuya & Muriel Calo (2016) Shifting gender roles: an analysis of violence against women in post-conflict Uganda. Development in Practice 26(3), 285–297. It connects to the statement on page 21 that “health services should be available to users in proportion to need (rather than social status)”.

Aren’t HBTQIA+ individuals a vulnerable group, due to the lack of appropriate health care overall and stigma (that may be heightened due to militarized masculinity in post-war settings)? Why were not these included? Or refugees displaced to other countries?

2) While I appreciate the clear distinctions of inequality vs. inequity, it seems that the authors sometimes use the terms interchangeably despite having stated the difference between the two terms. For example, on page 7 in this sentence: “We used the PROGRESS-Plus as an organizing framework to assess the structural determinants of health inequalities in post-conflict settings. Using PROGRESS-Plus enables the inclusion of a range of factors and circumvents reliance on any single measure to assess inequities, as this is insufficient.” It’s not always clear to me which of the concepts the authors actually want to explore. Relatedly, why were equality AND inequality included as search terms, but only equity and not inequity (according to Table S6)?

3) Some clarifications with respect to conflict vs. post-conflict are needed.

I appreciated the predetermined criteria for inclusion and exclusion. But I would add studies about ongoing conflict settings to the exclusion criteria if those were indeed excluded, in the parenthesis “(e.g., natural disasters/post-tsunami, terror victims, intimate partner violence)” on page 8.

The authors engage appropriately with the existing literature, for example by discussing the lack of consensus on what constitutes post-conflict. But does Colombia really count as a post-conflict setting? One of the studies included in analysis (Alzate, 2008) was published before the peace agreement between the government and FARC was signed in 2016, and since then the levels of violence have risen above pre-treaty levels. Several armed groups (including organized crime networks and the ELN guerrilla) are still actively fighting. What is the logic behind this inclusion?

Are there other studies included on active conflict settings at the time of publishing that count as post-war today?

5) While I commend the authors for providing such extensive documentation to support their arguments, the large number of tables and overlap between them rather reduce the level of clarity.

The authors have provided helpful documentation (such as the PRISMA check list and flow chart and the data extraction sheet) and describe clearly how quality was assessed. But is it necessary to have both Table S8 that summarize the findings, and S1 as a data extraction sheet? Both have summaries of the results, but these are not identical. Similarly, is it necessary to have three separate tables (S2–S4) sorted by data quality, when this information is also available in the data extraction sheet?

I also find the ordering/names of the tables a bit peculiar. They are mentioned in the following order in the text: S10, S5, S6, S1, S7, S2, S3, S4, S8, S9. Perhaps they could be renamed according to the order in which they are presented? It would make the process and document audit trail more transparent.

6) “Specifically, little is known about the effects of conflict and displacement on the provision of Sexual and Reproductive Health (SRH) services [90]” on page 20. I do not agree with this statement. A substantial amount of literature has been published on this topic in Conflict and Health, BMJ Global Health, and other outlets. See also the RAISE initiative at Columbia University, the BRANCH Consortium, and intervention studies on the Minimum Initial Service Package (MISP).

7) The section “Study identification” would be more logically placed before the section “Data extraction and analysis”, where the PRISMA chart is first mentioned. In fact, I do not see the point of having a separate heading for this section as it fits under “Study selection”.

8) It is interesting that, according to Table S7, Sub-Saharan Africa is substantially overrepresented in terms of geographical focus. Can this be reflected upon more directly? Does this reflect the incidence of post-conflict, or is it because of something else?

9) A page number seems to be missing from the quote reference on page 14.

Reviewer #2: Health inequality and or equity is an important area of studies while studying the same issue in the post conflict setting has an added value in the literature. The authors did a fantastic job to compile all the important study findings and did a systematic review on the subject area. This has provided a solid basis for identifying the research gap and future research in this area. The study findings and the conclusion is consistent and very strong in terms of the claim. However, due to my limitation on the study methodology I am a bit concerned about the authors' strong claim of non applicability of the meta analysis due to the heterogeneity of the study settings, populations and interventions. Someone from the relevant field should be approached for her/his opinion on this matter. Finally, I am wondering whether the authors came across this paper "Improving Maternal Health Care in a Post Conflict Setting: Evidence from Chittagong Hill Tracts of Bangladesh, The Journal of Development Studies (2018); https://doi.org/10.1080/00220388.2018.1554211 as it is not found in the reference or the list of studies covered.

Reviewer #3: SUMMARY OF REVIEWER

This review collates the current evidence on the structural determinants of health inequalities in post-conflict settings and describes the approaches used to address them. The methods and results were clearly described; however, the introduction and discussion should be re-structured to strengthen the study (and set up the importance/rationale of this review). Overall, this study will add important findings to the literature base on structural determinants of health in post-conflict settings.

INTRODUCTION

The authors have done a good job assessing the current evidence around health inequalities and conflict. However, the structure of the introduction felt slightly unclear and hard to follow. Sub-titles would help the reader follow the background narrative leading up to the rationale of the current study. Currently, the structure seems to be: inequalities persist globally and worsen in conflict; social determinants of health (SDH) in conflict; conflict in children; definition of inequalities vs inequities; definition of post-conflict settings; conflict as driver of health inequalities (ex DRC); right to health; impact of conflict on health and health systems documented but not on SDH; and jump to structural determinants of health inequalities in post-conflict settings. Review these paragraphs and find a structure that leads the reader to the gaps in the evidence and rationale of your study. For example, how are these inequalities specific to post-conflict settings and why is there less evidence on this area specifically. Start broad and then narrow on to the rationale of your review.

Specific points:

• The authors use the term ‘emerging countries.’ This would need to be defined or just stick with the term LMIC.

• The authors define inequalities vs inequities in health (page 3). The definition of social determinants of health, specifically structural determinants of health, is needed higher up in the introduction and its definition should be referenced. Social determinants of health and structural determinants of health inequalities seem to be use interchangeably. Define and stick to the one which you will use for the review.

• The outcome of interest (structural health inequalities) listed in methods needs to be defined as well (or instead of other combinations of those words).

• There is a lot of literature within the post-conflict setting on health. This felt like a summary of this was missing in the introduction.

METHODS

The overall design of the study was appropriate and adequate to answer the research question. The search strategy, inclusion and exclusion criteria and study selection were adequately described.

Specific points:

• Add reference for why these vulnerable groups were selected (see comment later re general population).

• Upon reading the methods, I assumed you would organise the results according to the PROGRESS-Plus framework for structural determinants. Could this be further integrated into each section of the results? For example, within the children sub-section, describing the main PROGRESS-Plus risk factors.

RESULTS

The results answer the research question and are well presented.

Specific points:

• It would be helpful to the reader to add percentages (n/N total publications; X%) within the results section. For example, in the sentence “The majority of the studies were based on data collected in sub-Saharan Africa (n/N; X%), followed by South Asia and the Middle East and North Africa region.”

• The general population was added as a vulnerable population group in the results but was not described in the methods. This should be added or explained. Furthermore, in this section, post-conflict was not mentioned. All results should reflect the specificities and complexities present in a post-conflict setting.

• IDP acronym is defined various times throughout the manuscript.

• The term and section entitled – Mentally Ill. Could this be re-framed as people with symptoms of mental illness (see for reference of bias free language - https://apastyle.apa.org/style-grammar-guidelines/bias-free-language/disability).

DISCUSSION

An important strength of this work is its focus on post-conflict settings. The results should be reviewed to clearly state how the findings relate to the specific constraints/circumstances of a post-conflict setting, as at times it felt like unclear or not explicitly stated. For example, the sentence on children in armed conflict does not connect to the importance of this findings on a post-conflict setting.

Specific points:

• The paragraph about children is about armed conflict. Could this be better integrated into the evidence from your review about post-conflict?

• Could sub-heading be added to the discussion? Again, this will help the reader follow the main findings. These could be based on sub-sections in the results section (population group and approaches to addressing health inequalities).

• A figure describing the recommendations for future research or summarising the gaps in the evidence base would be helpful to the readers.

6. PLOS authors have the option to publish the peer review history of their article (what does this mean?). If published, this will include your full peer review and any attached files.

Reviewer #1: No

Reviewer #2: **Yes: **Muhammad Badiuzzaman

Reviewer #3: No

---

## [Author Response · Author response to Decision Letter 0]

6 Nov 2021

Rebuttal letter

Response to Reviewers

PONE-D-21-08382

Health inequalities in post-conflict settings: A systematic review.

PLOS ONE

Dear Editor, 

We would like to thank you and the reviewers for evaluating our manuscript and providing constructive comments. We have carefully considered the points raised during the review process and have tried to address every one of them. We have copied the review decision from the submission menu of the editorial manager and used this format to write our responses. Below we provide the point-by-point responses to the comments. All modifications in the revised manuscript have been highlighted (with Track Changes). Of course, we remain at your disposal should you have any further suggestions for improvements of our paper.

Dear Dr. Bwirire,

Thank you for submitting your manuscript to PLOS ONE. After careful consideration, we feel that it has merit but does not fully meet PLOS ONE’s publication criteria as it currently stands. Therefore, we invite you to submit a revised version of the manuscript that addresses the points raised during the review process.

We look forward to receiving your revised manuscript.

Kind regards,

Jakob Pietschnig, PhD, MSc

Academic Editor

PLOS ONE

Journal Requirements:

>>> Author Response: Thank you very much for your constructive feedback! Accordingly, we went through the entire manuscript to ensure it meets PLOS ONE’s style requirements. Additionally, the file naming has been revised in line with the style template.

>>> Author Response: Thank you for pointing this out. We have done so accordingly. 

3. Please confirm that you have included all items recommended in the PRISMA checklist including: 

- a Supplemental file of the results of the individual components of the quality assessment, not just the overall score, for each study included. 

>>> Author response: As suggested by the reviewer, we have created a Supplemental file with the results of the individual components of the quality assessment for each study included. It is named S4 Table. Individual Quality Assessment for each study included

- details of reasons for study exclusions in the PRISMA flowchart and number of studies excluded for each reason.

>>> Author response: As suggested by the reviewer, we have provided the details of the reasons for study exclusions in the PRISMA flowchart and the number of studies excluded for each reason. 

- See https://journals.plos.org/plosmedicine/article?id=10.1371/journal.pmed.1000100#pmed-1000100-t003 for guidance on reporting.

Thank you.

>>> Author response: We have done so accordingly, following additional journal requirements kindly shared by the reviewer in the following article: “The PRISMA Statement for Reporting Systematic Reviews and Meta-Analyses of Studies that Evaluate Health Care Interventions: Explanation and Elaboration”

>>> Author response: Thank you for pointing this out. All extracted data that was used in this review are available in the predesigned data extraction sheet which is referred to in the supplementary material as S1 Table. Data extraction sheet information from selected studies.

5. Please remove all personal information, ensure that the data shared are in accordance with participant consent, and re-upload a fully anonymized data set. 

>>> Author response: Thank you for pointing this out. The data in Supplemental File (S1 Table) does not contain any personal information. 

6. Thank you for submitting the above manuscript to PLOS ONE. During our internal evaluation of the manuscript, we found significant text overlap between your submission and the following previously published works, some of which you are an author.

https://www.scribd.com/document/328300724/Estadisticas-de-Salud-Mundial

https://applications.emro.who.int/dsaf/dsa955.pdf

https://journals.plos.org/plosone/article?id=10.1371%2Fjournal.pone.0210071

https://www.who.int/social_determinants/links/events/conflicts_and_sdh_emro_revison_06_2007.pdf

https://apps.who.int/disasters/repo/8678.pdf

https://www.pure.ed.ac.uk/ws/files/19589731/Palmer_J._2014_._Changing_landscapes_changing_practice.pdf

https://www.coe.int/en/web/genderequality/what-is-gender-mainstreaming

https://onlinelibrary.wiley.com/doi/pdf/10.1111/tmi.12708

https://www.raiseinitiative.org/library/download.php?id=495

https://www.tandfonline.com/doi/abs/10.1080/17441692.2013.819587?journalCode=rgph20

Please revise the manuscript to rephrase the duplicated text, cite your sources, and provide details as to how the current manuscript advances on previous work. Please note that further consideration is dependent on the submission of a manuscript that addresses these concerns about the overlap in text with published work.

>>> Author response: Thank you for your comments and suggestions. We have done so accordingly.

Reviewers' comments:

Reviewer's Responses to Questions

Comments to the Author

1. Is the manuscript technically sound, and do the data support the conclusions?

Reviewer #1: Yes

Reviewer #2: Yes

Reviewer #3: Yes

2. Has the statistical analysis been performed appropriately and rigorously? 

Reviewer #1: Yes

Reviewer #2: N/A

Reviewer #3: Yes

3. Have the authors made all data underlying the findings in their manuscript fully available?

Reviewer #1: Yes

Reviewer #2: Yes

Reviewer #3: Yes

 4. Is the manuscript presented in an intelligible fashion and written in standard English?

Reviewer #1: Yes

Reviewer #2: Yes

Reviewer #3: Yes

5. Review Comments to the Author

Reviewer #1: Dear Editors,

Thank you for the opportunity to review this systematic review of research on health inequalities in post-conflict settings.

The article has many merits as it stands now. Relevant concepts (such as health inequalities, post-conflict, PROGRESS-Plus) are clearly defined. The aims of the study are appropriate, relevant, and narrow enough for the systematic review. My impression is that the authors conducted a rigorous, unbiased and comprehensive search for all studies according to the search strategy, which was clearly described and seems suitable for the aims.

Given the good shape of the article in its current condition, I believe the authors are capable of submitting an improved revised manuscript.

Response to Reviewer #1

Thank you very much for agreeing with us on the aim of our manuscript and recognizing that it has many merits as it stands now. Furthermore, we have read your comments carefully and tried our best to address them one by one by providing a detailed explanation of our way of working. By implementing your suggestions, as described below, we hope that you share our view that the manuscript has been improved after this revision. 

Some minor but important improvements are suggested below.

1) On page 6, the authors write “… the population of interest consisted of the most vulnerable population groups [34, 35] living in post-conflict areas such as women, children, internally displaced persons (IDPs), and the mentally ill – as these are generally affected the most.” Why were these groups identified as most vulnerable? In the results section it appears as if this choice was data-driven (e.g. “It is widely recognized that armed conflict has a particularly negative impact on women's lives [105, 106]” on page 13 and the paragraph starting with “Women and men have different needs, living conditions, and circumstances, including unequal access to and control over power, resources, human rights, and institutions, including the justice system” on page 21). But the earlier statement on page 6 makes it sound like it is theory-driven. Some clarification as to what guided the sub-group selection would be helpful.

>>> Author response: Thank you very much for the great question. Indeed, it might be difficult for the reader to understand what guided the sub-group selection during this review. Putting the sub-group selection process within the broad context of our review, we aimed to collate the current knowledge and knowledge gaps about structural determinants of health inequalities in post-conflict settings. Our understanding is that it is important to identify the characteristics of post-conflict environments that create vulnerability because they either create or contribute towards losses, which lead to vulnerability and negative well-being. Indeed, beforehand the majority of the most vulnerable population sub-groups were already identified. These included Women, Children, IDPs, and People with symptoms of mental symptoms. While applying a stepwise data extraction approach, we found an additional sub-group of interest which consisted of the general population and added this afterward to the final sub-group selection. To help the reader, we have slightly changed the text on page 6 as follows: 

“… the population of interest consisted of the most vulnerable population groups [34, 35] living in post-conflict areas such as the general population, women, children, internally displaced persons (IDPs), and the people with symptoms of mental illness – as these are generally affected the most.”

I agree children, IPDs and the mentally ill are likely particularly vulnerable, but why are women included? Since women may make up far more than 50 percent of the population in post-conflict settings due to high male battle-related mortality, what is the logic behind identifying them as one of the most vulnerable groups? Feminist security scholars such as Cynthia Enloe have criticized the merged category “womenandchildren” for removing women’s agency by only portraying them as mindless victims. While I do not necessarily disagree that women are a vulnerable group, a clearer explanation to this effect would help.

>>> Author response: Thank you very much for agreeing with us that children, IDPs, and people with symptoms of mental illness are particularly vulnerable in this environment and therefore should be part of the sub-group selection. As noted previously, the final sub-group selection was based on a combination of both: a beforehand selection of the most sub-groups (including women) and an afterward selection of the general population sub-group. 

While we tend to agree with your comment about the Feminist security scholars who have criticized the merged category “women and children” for removing women’s agency by only portraying them as mindless victims. We would not feel comfortable discussing the most vulnerable sub-groups in a post-conflict environment without giving proper attention to women and we have addressed this in the revised version of the manuscript: “In particular, we would not feel comfortable discussing the most vulnerable sub-groups in this environment without giving proper attention to women. The reasons for this are multifold: 1) traditional networks that form the foundation of women’s social and family roles may be disrupted during the conflict, 2) women may also leave their communities and husbands for other partners to guarantee a source of food for themselves and their families, and 3) women have to assume the role of the primary breadwinners and caretakers of their families – putting them more at risk [37, 38].” For these reasons, we prefer to point the reader in this direction and maintain the women sub-group in the most vulnerable study population group.

A specific focus on women also risks sidelining men from post-conflict interventions, as argued on page 21. The study from Uganda [86] suggests that women were more likely to be widowed, which also likely means that men are more likely to die in war. Is that not also a vulnerability? Additionally, past research has shown that if post-war aid only focuses on women, it may drive intimate partner violence perpetrated by men who have lost their socioeconomic position as main breadwinners. See Sengupta, Anasuya & Muriel Calo (2016) Shifting gender roles: an analysis of violence against women in post-conflict Uganda. Development in Practice 26 (3), 285–297. It connects to the statement on page 21 that “health services should be available to users in proportion to need (rather than social status)”.

>>> Author response: Thank you very much for your constructive feedback! Although we can agree in general with the reviewer’s assessment that both women and men are vulnerable in this environment, we still want to explain why women are most vulnerable than men in post-conflict settings. First, women and men have different exposures to situations that affect their health and access to health care, they also have differential power to influence decisions regarding the provision of health services. In particular, Sorenson (1998) documented one feature of post-conflict environments that is often ignored both by academics as well as policymakers which is the situation of women. Second, in environments where the involvement of women in the labor market is not the norm, women either remain unemployed or become underemployed. According to Bodewig (2002), because women may not be exceptionally educated, even the employment opportunities they could avail would be low-paying jobs. 

We finally want to clarify that by giving a bit more attention to women's needs in this environment, our intention was not to take men’s needs away from post-conflict interventions. We only want to specify that women have different roles and perspectives from those men have.

The involvement of women and men at all stages in a response to post-conflict situations is desirable and beneficial.

Aren’t HBTQIA+ individuals a vulnerable group, due to the lack of appropriate health care overall and stigma (that may be heightened due to militarized masculinity in post-war settings)? Why were not these included? Or refugees displaced to other countries?

>>> Author response: Thank you very much for these great questions. The reviewer is most likely referring to the LGBTQ+ (which stands for Lesbian, Gay, Bisexual, Transgender, and Queer) community. Here again, we can agree with the reviewer’s assessment that both the LGBTQ+ community as well as refugees are vulnerable populations in post-conflict environments.

As for the LGBTQ+ community, it was not specified beforehand as one of the most vulnerable sub-groups in this environment. Because of the already existing “LGBTQ discrimination”, their identification (based on sexual orientation or gender identity) would not be easy in practice and maybe could reinforce discrimination that is already present. For this reason, we believe it could have been hard to tell something about this sub-group based on currently available data. However, we certainly believe more attention should be paid to structural inequalities that keep the LGBTQ+ community apart, and that has been highlighted in post-conflict settings. For now, we prefer to point the reader in the direction of the subgroups we have identified as being most vulnerable in this environment (as explained above).

Regarding refugees, we have presented information on page 3 of our manuscript as follows: “For example, children and young people often comprise the highest proportion of the population in refugee and internally displaced persons (IDP) camps and are exposed to risks over which they have little control [6, 7].” By doing so, we acknowledge that it is a vulnerable group but not the most vulnerable one. On the other hand, when people cross an international border to find safety during a crisis, they come under the protection of the UNHCR. That means they can access the funding and resources of the UN. Decades of global news coverage have shown that life for refugees is difficult. But they do have some protections, often have access to camps built to provide shelter and food. However, we were ultimately interested in the Internally Displaced Persons (IDPs), as this sub-group stays within their own country and remains under the protection of their government, even if that government is the reason for their displacement. When people flee home but don’t cross an international border to find safety, they remain under the protection of their own country. It may seem like a better situation to find a temporary home within your country of origin. But in reality, it’s often a much more difficult situation. A country at war has extremely limited resources to help displaced people. And internally displaced persons are not protected under the UN. IDPs have not crossed the border to find safety and are therefore amongst the most vulnerable in the world because they often move to areas where it is difficult to deliver humanitarian assistance. Often, they are sheltered among the poorest communities, sharing scarce resources with host communities for years. Therefore, it can be an added value to make a distinction in how to respond to the needs of refugees or internally displaced people (IDPs), and provide critical services to these groups in vulnerable situations to ensure they can live with the safety and dignity they deserve.

2) While I appreciate the clear distinctions of inequality vs. inequity, it seems that the authors sometimes use the terms interchangeably despite having stated the difference between the two terms. For example, on page 7 in this sentence: “We used the PROGRESS-Plus as an organizing framework to assess the structural determinants of health inequalities in post-conflict settings. Using PROGRESS-Plus enables the inclusion of a range of factors and circumvents reliance on any single measure to assess inequities, as this is insufficient.” It’s not always clear to me which of the concepts the authors actually want to explore. Relatedly, why were equality AND inequality included as search terms, but only equity and not inequity (according to Table S6)?

>>> Author response: Thank you very much for appreciating our efforts to explain both terms: inequality and inequity. Indeed, by the fact that in languages (including English) where the two terms exist, they are often used interchangeably. Generally, the technical distinction between the two concepts of health inequality and health inequity is that health inequality can refer to all differences, and health inequity refers just to those differences which are unnecessary, avoidable, unfair, and unjust. To avoid confusion, we have provided the meaning of each concept. Accordingly, we have revised both concepts throughout our manuscript to help the reader better understand which of the two concepts the authors want to explore. 

Finally, inequity was not included as a separate key concept and search term because it refers to a specific type of health inequality that denotes an unjust difference in health. As mentioned previously, this aspect is covered by health inequality as this concept refers to all differences?

Per supplemental Table 6. health inequality was included in the key concepts and search terms for our review. To mitigate the risk of whether we might have missed certain evidence, a significant effort was made by applying a rigorous search strategy (including the use of ineq*) and identifying articles from scientific and professional databases only (Page 29 of the revised manuscript). 

3) Some clarifications with respect to conflict vs. post-conflict are needed.

I appreciated the predetermined criteria for inclusion and exclusion. But I would add studies about ongoing conflict settings to the exclusion criteria if those were indeed excluded, in the parenthesis “(e.g., natural disasters/post-tsunami, terror victims, intimate partner violence)” on page 8.

>>> Author response: Thank you very much for your constructive feedback. We have addressed your comments as described below: “Abstract-only studies, conference abstracts, studies about ongoing conflict settings, and studies on serving or former military personnel were excluded.”

The authors engage appropriately with the existing literature, for example by discussing the lack of consensus on what constitutes post-conflict. But does Colombia really count as a post-conflict setting? One of the studies included in analysis (Alzate, 2008) was published before the peace agreement between the government and FARC was signed in 2016, and since then the levels of violence have risen above pre-treaty levels. Several armed groups (including organized crime networks and the ELN guerrilla) are still actively fighting. What is the logic behind this inclusion?

>>> Author response: Thank you very much for this great question and for agreeing with us on the efforts to clarify what constitutes a post-conflict setting. As explained on page 4 of our manuscript, there is no consensus in the literature about the definition of post-conflict settings. Post-conflict describes the period immediately after a conflict is over. But, the term is more difficult to define for many reasons. Firstly, conflicts do not necessarily end with the signing of official peace agreements. Secondly, low-intensity conflict replaces violent conflict because one or several actors are either excluded or not content with the peace agreement. The end of a conflict and subsequently the start of the post-conflict phase, therefore, become difficult to determine. For these reasons, we have elaborated on what we understand under post-conflict settings and, we have also described what characterizes this environment. Specifically for Colombia, this country is a good example of a post-conflict situation. Although the peace agreement was not signed yet, the other characteristics of a post-conflict setting were present (as described on Page 5 of the revised manuscript). 

The final selection of Alzate’s paper (The sexual and reproductive rights of internally displaced women: the embodiment of Colombia's crisis) was based on the fact that this paper met all criteria to be included in the final review. The fact that all of the post-conflict characteristics were not present simultaneously was not an issue as – at the time of the final paper selection, Colombia was at a stage of recovery from conflict or crisis and a stage of rebuilding and reconstruction starting from emergency and stabilization followed by transition and recovery, and peace and development.

Are there other studies included on active conflict settings at the time of publishing that count as post-war today?

>>> Author response: Thank you for pointing this out. In line with the explanation above, studies about structural determinants of health inequalities in contexts other than post-conflict environments were excluded.

5) While I commend the authors for providing such extensive documentation to support their arguments, the large number of tables and overlap between them rather reduce the level of clarity.

The authors have provided helpful documentation (such as the PRISMA check list and flow chart and the data extraction sheet) and describe clearly how quality was assessed. But is it necessary to have both Table S8 that summarize the findings, and S1 as a data extraction sheet? Both have summaries of the results, but these are not identical. Similarly, is it necessary to have three separate tables (S2–S4) sorted by data quality, when this information is also available in the data extraction sheet?

>>> Author response: We thank the reviewer for the nice comments and suggestions. To increase the level of clarity, we have decreased the number of tables as follows:

We combined Tables S1 (Data Extraction Sheet) and S8 (Summary of study findings) into one table (New S1). Importantly, we moved the findings S8 to the new S1 Table to make sure the reported findings are identical. Finally, tables (S2–S4) have also been combined to cut down the number of tables. 

I also find the ordering/names of the tables a bit peculiar. They are mentioned in the following order in the text: S10, S5, S6, S1, S7, S2, S3, S4, S8, S9. Perhaps they could be renamed according to the order in which they are presented? It would make the process and document audit trail more transparent.

>>> Author response: Thank you for your nice reminder. We revised the ordering and naming of the tables throughout the manuscript to make them clearer and have the numbering in line with the order in which they are mentioned in the manuscript. 

6) “Specifically, little is known about the effects of conflict and displacement on the provision of Sexual and Reproductive Health (SRH) services [90]” on page 20. I do not agree with this statement. A substantial amount of literature has been published on this topic in Conflict and Health, BMJ Global Health, and other outlets. See also the RAISE initiative at Columbia University, the BRANCH Consortium, and intervention studies on the Minimum Initial Service Package (MISP).

>>> Author response: Thank you very much for your constructive feedback. We agree with you that more is currently known about the effects of conflict and displacement on the provision of Sexual and Reproductive Health (SRH) services in post-conflict settings. For the sake of clarity, we have rephrased this in the revised version of the manuscript: “Although a substantial amount of literature has been published on the effects of conflict and displacement on the provision of Sexual and Reproductive Health (SRH) services in post-conflict settings, there is a lot we do not know yet about how to address the problem. These findings concur with the conclusion of a recent systematic review of monitoring and evaluation indicators for SRH, showing that rigorous reporting on a core list of indicators is a prerequisite for identifying the changes that a country wants to see concerning SRH and the investments required to save lives [111].” 

7) The section “Study identification” would be more logically placed before the section “Data extraction and analysis”, where the PRISMA chart is first mentioned. In fact, I do not see the point of having a separate heading for this section as it fits under “Study selection”.

>>> Author response: Thank you very much for your suggestion that helped us to improve this manuscript. This section has been revised accordingly.

8) It is interesting that, according to Table S7, Sub-Saharan Africa is substantially overrepresented in terms of geographical focus. Can this be reflected upon more directly? Does this reflect the incidence of post-conflict, or is it because of something else?

>>> Author response: Thank you very much for this great question. In our opinion, the fact that 27 out of 63 (42.8%) selected studies were conducted in Sub-Saharan Africa (as described on page 13) is an interesting finding on its own. Indeed, it could reflect a high incidence of post-conflict situations in this specific region. In their study on Violent Conflict and Its Impact on Health Indicators in sub- Saharan Africa, David R. Davis & J. Kuritsky (2002) documented high levels of conflict in Sub-Saharan African countries but also poor health indicators. Furthermore, this could mean that Sub-Saharan Africa can be used to inform post-conflict planning when it involves determining the potential geographic extent strategy targeting health inequality-related questions. 

9) A page number seems to be missing from the quote reference on page 14.

>>> Author response: Thank you very much for the comment. We have added a page number to help the reader immediately locate the source of the quote referenced on Page 14 from the original article.

Reviewer #2: Health inequality and or equity is an important area of studies while studying the same issue in the post-conflict setting has an added value in the literature. The authors did a fantastic job to compile all the important study findings and did a systematic review on the subject area. This has provided a solid basis for identifying the research gap and future research in this area. The study findings and the conclusion is consistent and very strong in terms of the claim. However, due to my limitation on the study methodology I am a bit concerned about the authors' strong claim of non applicability of the meta analysis due to the heterogeneity of the study settings, populations and interventions. Someone from the relevant field should be approached for her/his opinion on this matter. Finally, I am wondering whether the authors came across this paper "Improving Maternal Health Care in a Post Conflict Setting: Evidence from Chittagong Hill Tracts of Bangladesh, The Journal of Development Studies (2018); https://doi.org/10.1080/00220388.2018.1554211 as it is not found in the reference or the list of studies covered.

Response to Reviewer #2

>>> Author response: We thank the reviewer for the positive feedback on our manuscript and for the suggestion on how we could further improve it. Although we understand the reviewer’s question about the methodological choice for not using meta-analysis in this review, we want to ensure the reviewer that we explored this aspect more in detail. As reported in the manuscript (Pg 9), before making the final decision on how to proceed “. …we contacted experts for advice (e.g., the plan to use PRISMA standard in combination with PRISMA-E was discussed with the editor in chief of the Campbell Collaboration Library)”.

For several reasons, we strongly believe that using qualitative approaches was appropriate to report on this systematic review:

• The approach allowed to describe, analyze, criticize, and summarize data by the vulnerable sub-groups, 

• The qualitative inferences were presented as systematically as possible as part of the evidence in this systematic review.

• The study strengths and limitation section has a section explaining this as a limitation and more importantly how the authors have dealt with it. 

Our choice to focus on the qualitative approaches was the most appropriate because the data available is so heterogeneous that a meta-analysis is impossible. If we would run the study again, we still would not choose a meta-analysis because of this heterogeneity. More importantly, we extracted the available data judiciously and purposively with an eye to what is relevant for our main review questions.

We finally thank the reviewer for helping us with important references: "Improving Maternal Health Care in a Post Conflict Setting: Evidence from Chittagong Hill Tracts of Bangladesh, The Journal of Development Studies (2018); https://doi.org/10.1080/00220388.2018.1554211

Although this important study was published during the timing of our data extraction, it was not included in our final selection for this review because the main focus of this study is not about inequality, the study evaluates a development program with an important maternal health care component in the Chittagong Hill Tracts of Bangladesh - a post-conflict setting. 

We are currently working on the next manuscript on health inequality trends with a specific focus on Maternal Health Care – we will use this important reference to address any relevant questions.

Reviewer #3: SUMMARY OF REVIEWER

This review collates the current evidence on the structural determinants of health inequalities in post-conflict settings and describes the approaches used to address them. The methods and results were clearly described; however, the introduction and discussion should be re-structured to strengthen the study (and set up the importance/rationale of this review). Overall, this study will add important findings to the literature base on structural determinants of health in post-conflict settings.

INTRODUCTION

The authors have done a good job assessing the current evidence around health inequalities and conflict. However, the structure of the introduction felt slightly unclear and hard to follow. Sub-titles would help the reader follow the background narrative leading up to the rationale of the current study. Currently, the structure seems to be: inequalities persist globally and worsen in conflict; social determinants of health (SDH) in conflict; conflict in children; definition of inequalities vs inequities; definition of post-conflict settings; conflict as driver of health inequalities (ex DRC); right to health; impact of conflict on health and health systems documented but not on SDH; and jump to structural determinants of health inequalities in post-conflict settings. Review these paragraphs and find a structure that leads the reader to the gaps in the evidence and rationale of your study. For example, how are these inequalities specific to post-conflict settings and why is there less evidence on this area specifically. Start broad and then narrow on to the rationale of your review.

>>> Author response: We thank the reviewer for the suggested modifications in the introduction section. To lead the reader to the gaps in the evidence and rationale of our study, we revised the introduction section (from pg.3-8) as follows:

1. Sub-titles were created to improve the structure: 

• Social determinants versus Structural determinants of health

• Health inequalities versus health inequities

• Conflict versus post-conflict settings

• Right to health

2. Paragraphs were moved in the background narrative leading up to the rationale of the current study.

3. Unnecessary details are removed from this section.

Specific points:

• The authors use the term ‘emerging countries.’ This would need to be defined or just stick with the term LMIC.

>>> Author response: Thank you for your comment and suggestion. we have rephrased this in the revised version of the manuscript: “…the social determinants affecting children in crisis settings are the same as those affecting children in Low and Middle-Income countries….”

• The authors define inequalities vs inequities in health (page 3). The definition of social determinants of health, specifically structural determinants of health, is needed higher up in the introduction and its definition should be referenced. Social determinants of health and structural determinants of health inequalities seem to be used interchangeably. Define and stick to the one which you will use for the review.

>>> Author response: Thank you very much for your constructive feedback. We have addressed your comments as described below: 

“Social determinants versus Structural determinants of health

Because the concept of ‘social determinants’ refers simultaneously to both the determinants of health and the determinants of inequalities in health, we will expand our review of social determinants of health to include the terms ‘‘structural determinants of health’’ as we assess the literature on this topic. In this article, the term "structural determinants of health inequalities" refers specifically to the interplay between the socio-economic-political context, structural mechanisms generating social stratification, and the resulting socio-economic position of individuals. These mechanisms configure social groups' health opportunities based on their placement within hierarchies of power, prestige, and access to resources [9]. We hypothesize that structural determinants affect the distribution of resources and have the potential to influence mental health inequalities.”

• The outcome of interest (structural health inequalities) listed in methods needs to be defined as well (or instead of other combinations of those words).

>>> Author response: Thank you very much for these comments. We agree with the reviewer that the outcome of interest (structural health inequalities) should be defined as this is required to interpret the results. As noted above, the term "structural determinants of health inequalities" refers specifically to the interplay between the socio-economic-political context, structural mechanisms generating social stratification, and the resulting socio-economic position of individuals. These mechanisms configure social groups' health opportunities based on their placement within hierarchies of power, prestige, and access to resources [9]. We hypothesize that structural determinants affect the distribution of resources and have the potential to influence mental health inequalities.

• There is a lot of literature within the post-conflict setting on health. This felt like a summary of this was missing in the introduction.

>>> Author response: Thank you for emphasizing the need to clarify that there is already a lot of literature available on health within the post-conflict setting. As suggested, we have summarized this in the introduction (see page pg.3-8). Additionally, in reply to your main comment, we have restructured the introduction section to address your comments.

METHODS

The overall design of the study was appropriate and adequate to answer the research question. The search strategy, inclusion and exclusion criteria and study selection were adequately described.

Specific points:

• Add reference for why these vulnerable groups were selected (see comment later re general population).

>>> Author response: Thank you for this helpful suggestion. We have added references for why these vulnerable sub-groups were selected. To help the reader, we have slightly changed the text on page 7 of the manuscript as follows: “… the population of interest consisted of the most vulnerable population groups [34, 35] living in post-conflict areas such as the general population, women, children, internally displaced persons (IDPs), and the people with symptoms of mental illness – as these are generally affected the most.”, they were selected because, often, they have been referred to in the literature as most vulnerable in post-conflict environments.

• Upon reading the methods, I assumed you would organise the results according to the PROGRESS-Plus framework for structural determinants. Could this be further integrated into each section of the results? For example, within the children sub-section, describing the main PROGRESS-Plus risk factors.

>>> Author response: We thank the reviewer for this valid comment and the suggestions. The reviewer is right, it could have been possible (an option) to analyze our data using the PROGRESS-Plus framework for structural determinants on health inequalities in post-conflict settings. However, we initially discussed the way they would use this framework to guide/support this review. In the manuscript (Pg 8), we referred to the framework under search strategy, underlining the fact that it enables the inclusion of a range of factors needed to measure inequities. According to Jennifer O’Neill et al. (2013), the PROGRESS-PLUS acronym can be used as an aide-mémoire, a framework to guide data extraction. We used all PROGRESS-PLUS acronyms to make the best selection of the studies to be included in systematic reviews. To ensure that decisions about structural determinants of health inequalities should be targeted with any future intervention, we found it useful to involve the most vulnerable groups – therefore, the results have been organized according to the representatives from these sub-groups.

Although there could be some differences in the way these results were presented (PROGRESS-PLUS framework versus the most vulnerable sub-groups), these results depend entirely on how the studies were selected and if the quality assessment was done. In both situations, we believe the results are useful as they contribute to the body of knowledge on structural determinants of health inequalities in post-conflict settings.

RESULTS

The results answer the research question and are well presented.

Specific points:

• It would be helpful to the reader to add percentages (n/N total publications; X%) within the results section. For example, in the sentence “The majority of the studies were based on data collected in sub-Saharan Africa (n/N; X%), followed by South Asia and the Middle East and North Africa region.”

>>> Author response: Thank you very much for the helpful suggestions. We have integrated them in the results section (pg.13) as follows: “The majority of the studies were based on data collected in sub-Saharan Africa (27/63 total publications; 42%), followed by South Asia (7/63 total publications; 11%) and the Middle East (6/63 total publications; 9%) and North Africa region (6/63 total publications; 9%). Three regions, including Asia Pacific, South America, and Europe, had similar data - This is potentially an indicator that some recent peace agreements have been reached in these regions. Only one study collected data in the Russia and Eurasia region.”

• The general population was added as a vulnerable population group in the results but was not described in the methods. This should be added or explained. Furthermore, in this section, post-conflict was not mentioned. All results should reflect the specificities and complexities present in a post-conflict setting.

>>> Author response: Thank you for the helpful suggestions. We have integrated them in the methods section (pg.7) as follows: “The outcome of interest was structural health inequalities and the population of interest consisted of the most vulnerable population groups [35, 36] living in post-conflict areas such as the general population, women, children, IDPs, and people with symptoms of mental illness – as these are generally affected the most, they were selected because, often, they have been referred to in the literature as most vulnerable in post-conflict environments.” On several occasions, we have also made reference to post-conflict areas in the Materials and methods sections and sub-headings (from pages 7 to 12).

• IDP acronym is defined various times throughout the manuscript.

>>> Author response: Thank you very much for pointing this out. We revised this throughout the manuscript and used the IDP acronym appropriately.

• The term and section entitled – Mentally Ill. Could this be re-framed as people with symptoms of mental illness (see for reference of bias free language - https://apastyle.apa.org/style-grammar-guidelines/bias-free-language/disability).

>>> Author response: Thank you for your comments. As suggested by the reviewer, we have re-framed the term Mentally Ill as people with symptoms of mental illness throughout the manuscript.

DISCUSSION

An important strength of this work is its focus on post-conflict settings. The results should be reviewed to clearly state how the findings relate to the specific constraints/circumstances of a post-conflict setting, as at times it felt like unclear or not explicitly stated. For example, the sentence on children in armed conflict does not connect to the importance of this findings on a post-conflict setting.

Specific points:

• The paragraph about children is about armed conflict. Could this be better integrated into the evidence from your review about post-conflict?

>>> Author response: Thank you for your nice reminder. We revised the paragraph about children to make it clearer that it is more about post-conflict settings than conflict situations. The revised text reads as follows:

“Children

The effects of armed conflict on children’s physical health and development are not well understood. In particular, little is known about how the circumstances of a post-conflict setting could affect the health status of children. A recent systematic review of the effects of armed conflict on child health and development [8] shows some gaps in the medium and long-term effects. The authors report a striking paucity of data on child health in relation to conflict and further supports the need for research on how conflict and structural inequalities affect children and subsequent generations. According to the Human Security Centre, child mortality is high in post-conflict countries due to food insecurity leading to malnutrition and the disruption of the water supply and sanitation.[118]. This suggests that children are more vulnerable during and after conflicts, as a consequence of a quick transition from conflict to post-conflict in these environments”.

• Could sub-heading be added to the discussion? Again, this will help the reader follow the main findings. These could be based on sub-sections in the results section (population group and approaches to addressing health inequalities).

>>> Author response: Many thanks for the helpful suggestions. To help the reader follow the main findings of our review, we have integrated sub-headings in the discussion section of the manuscript (from pg.22-27) as follows: general population, Women, Children, IDPs, and Persons with symptoms of mental illness) and approaches to addressing health inequalities.

• A figure describing the recommendations for future research or summarising the gaps in the evidence base would be helpful to the readers.

>>> Author response: Although we are happy to add figures, we are not sure whether a visual representation would be the best way to present the recommendations and/or summarize the gaps. In fact, we are not even sure how such a figure would look like. If the reviewer has clear suggestions on this, then we are happy to reconsider.

 6. PLOS authors have the option to publish the peer review history of their article (what does this mean?). If published, this will include your full peer review and any attached files.

>>> Author response: Thanks for your comment. We agree with you that publishing the peer review history increases transparency and accountability and helps to reinforce the validity of our research – it is good practice.

Do you want your identity to be public for this peer review? For information about this choice, including consent withdrawal, please see our Privacy Policy.

Reviewer #1: No

Reviewer #2: Yes: Muhammad Badiuzzaman

Reviewer #3: No

---

## [Decision Letter · Decision Letter 1]

1 Dec 2021

PONE-D-21-08382R1Health inequalities in post-conflict settings: A systematic review.PLOS ONE

Dear Dr. Bwirire,

Thank you for submitting your manuscript to PLOS ONE. After careful consideration, we feel that it has merit but does not fully meet PLOS ONE’s publication criteria as it currently stands. Therefore, we invite you to submit a revised version of the manuscript that addresses the points raised during the review process.

I realize that it is uncommon to receive a major revision decision after having received a minor revision decision in the first turnover. However, the novel points raised by Reviewer 1 particularly in regard to the apparently copied sentences from other works (see Top comment on page 2 of the review from Reviewer 1), are a reason for substantial concerns. I encourage you to address all points that Reviewer 1 has raised and convincingly justify if you do not follow their concern in certain places. Moreover, I expect you to carefully recheck the entire ms. for wordings that are not original and reword them to represent original contributions instead of copied content.

We look forward to receiving your revised manuscript.

Kind regards,

Jakob Pietschnig, PhD, MSc

Academic Editor

PLOS ONE

Reviewers' comments:

Reviewer's Responses to Questions

**Comments to the Author**

1. If the authors have adequately addressed your comments raised in a previous round of review and you feel that this manuscript is now acceptable for publication, you may indicate that here to bypass the “Comments to the Author” section, enter your conflict of interest statement in the “Confidential to Editor” section, and submit your "Accept" recommendation.

Reviewer #1: (No Response)

Reviewer #2: All comments have been addressed

2. Is the manuscript technically sound, and do the data support the conclusions?

Reviewer #1: No

Reviewer #2: Yes

3. Has the statistical analysis been performed appropriately and rigorously? 

Reviewer #1: No

Reviewer #2: N/A

4. Have the authors made all data underlying the findings in their manuscript fully available?

Reviewer #1: Yes

Reviewer #2: Yes

5. Is the manuscript presented in an intelligible fashion and written in standard English?

Reviewer #1: Yes

Reviewer #2: Yes

6. Review Comments to the Author

Reviewer #1: (No Response)

Reviewer #2: I have gone through your revised manuscript. You have addressed my review comments and concerns. The revised version looks quite strong and impressive.

7. PLOS authors have the option to publish the peer review history of their article (what does this mean?). If published, this will include your full peer review and any attached files.

Reviewer #1: No

Reviewer #2: No

---

## [Author Response · Author response to Decision Letter 1]

7 Jan 2022

Rebuttal letter

Response to Reviewer(s)

PONE-D-21-08382R1

Health inequalities in post-conflict settings: A systematic review.

PLOS ONE

Dear Editor, 

We would like to thank you and the reviewers for evaluating our manuscript and providing additional constructive comments. We appreciate your input and we have tried to address your suggestions about the reviewer’s comment as best as we could. We have copied the review decision from the submission menu of the editorial manager and used this format to write our responses. Below we provide the point-by-point responses to the comments. We carefully rechecked the entire manuscript for wordings that were not original and reworded them to represent original contributions. All modifications in the revised manuscript have been highlighted (with Track Changes). Of course, we remain at your disposal should you have any further suggestions for improvements to our paper.

Dear Editors,

Thank you for inviting me to review this manuscript a second time. I believe the authors have done a decent job addressing the comments by all reviewers in the first round of revision, but several major issues still remain before the manuscript is ready for publication.

* Why did you not do a database search to cover the period December 2020 and onwards? This would add to the longevity of the article, as it is already missing a whole year of potentially relevant publications for inclusion in the systematic review. Limiting the new search to the past 12 months should likely not be too time consuming.

>>> Author Response: Thank you for this very important insight into the design and conduct of this review. As explained in the search strategy (Page 7), the database search was a comprehensive search initiated in March 2018 and ultimately updated in December 2020. Thereafter, all selected papers were reviewed and our initial submission to PLoS One took place in March 2021. Hence, the year is mostly caused by the review time. That being said, we consulted PLOS ONE additional guidance (Sagoo GS et al., March 3, 2009) to critically look at the methodology. Not more than two years have passed between the searches and submission. However, we would be happy to update the database search if the reviewer would strongly feel the need for it, but this would require an additional extension.

* Even if the review is supposed to be restricted to post-conflict, there are some places where it sounds as if conflict is actually the main interest. The first 1.5 pages only covers conflict. The same goes for the paragraph “Many studies have investigated --- not well documented” on page 6. Another place is page 13 where the authors write “We found thirty studies --- health in conflict-affected general populations”. Why did you not write about post-conflict, if this is the context you are interested in?

>>> Author Response: Thank you for emphasizing the need to clarify the context we are interested in. Because the end of a conflict and the start of the post-conflict phase is difficult to determine, we have previously elaborated on what we understand under post-conflict settings. As explained on page 6 of our manuscript, post-conflict describes the period immediately after a conflict is over. But, the term is more difficult to define for many reasons. Firstly, conflicts do not necessarily end with the signing of official peace agreements. Secondly, low-intensity conflict replaces violent conflict because one or several actors are either excluded or not content with the peace agreement. The end of a conflict and subsequently the start of the post-conflict phase, therefore, become difficult to determine. 

At this point, we still want to confirm that we are interested in a post-conflict setting for many reasons:

• this is indicated in the title of our manuscript: “ Health inequalities in post-conflict settings: A systematic review” – Page 1

• this systematic review aims to collate the current knowledge and identify knowledge gaps about structural determinants of health inequalities within post-conflict settings and assess the effects of approaches aimed at addressing structural determinants of health inequalities in this environment- Page 6

• the search strategy was comprehensive to allow for a wide variety of study designs and interpretations of structural determinants of health inequalities in post-conflict environments to be included – Pages 6-7

However, we clearly understood that additional clarifications were needed to orient the reader in a post-conflict-focused systematic review. By explicitly referring to conflict and post-conflict environments instead of conflict only, we rephrased the following in the manuscript:

• Conflict and post-conflict environments are now referred to in the first 1.5 pages instead of conflict only. 

• On page 6: The same goes for the paragraph “Many studies have investigated --- not well documented”. 

• As regards the term ‘conflict-affected’: we use this term because, obviously, the interest in post-conflict areas arises from the very fact that the conflict-affected the health situation of the population.

• On page 14: the text reads as follows “We found thirty studies --- health in conflict and post-conflict environments”. 

* The introduction has some structural problems. It takes almost six pages before the authors explain what the article is about. I propose moving the paragraph “Very few empirical studies --- health inequalities in post-conflict settings” on pages 6-7 to the very beginning of the article.

>>> Author Response: Thank you very much for your feedback! As the reviewer suggested, we have moved the section starting with “Very few empirical studies --- health inequalities in post-conflict settings” on pages 6-7 to the very beginning of the article. We added the following paragraph at the first page of the Introduction:

” Very few empirical studies have measured health equity in conflict-affected settings, but those have suggested that conflict has a significant impact [4]. Most importantly, there is limited evidence on appropriate approaches for health equity in such environments. To accomplish equity, the prerequisite is first to ascertain the presence of inequality so that targeted interventions can be planned, implemented, and strategically monitored to determine the progress [5]. This systematic review aims to collate the current knowledge and identify knowledge gaps about structural determinants of health inequalities within post-conflict settings and assess the effects of approaches aimed at addressing structural determinants of health inequalities in this environment. It contributes to the body of knowledge on structural determinants of health inequalities in post-conflict settings which is also the main outcome of this study”.

We hope this helps to immediately indicate to the reader what the article is about.

* The subheadings “Social determinants vs structural determinants of health”, “Conflict versus post-conflict settings”, and “Right to health” are useful, but these sections would be more appropriately placed in a section called, for example, “Key concepts” or “Background”. This section should come after the introduction, where the authors should explain what they are about to do.

>>> Author Response: Thank you for your comment. We followed the instructions provided by the reviewer and created a new section after the introduction. This section is named: Key concepts. To explain to the reader what we main by key concepts, we added the following paragraph at the beginning of this new section as follows:

” To study health inequalities, it is of importance to clarify the following concepts: social determinants versus structural determinants of health; inequalities versus inequities in health; conflict versus post-conflict settings, effects of conflict on health equity and right to health is relevant. Without clearly understanding those key concepts, no one can develop effective policies for health equity. This paper helps to define those key questions that are needed to better understand health inequalities in post-conflict settings, such as...”

* There should be a new subheading after “Right to health” when the authors go back to explaining their study (the sentence starting with “Many studies have investigated…” on page 6).

>>> Author Response: Thank you very much for your suggestion that helped us to improve the structure of the introduction section. We added a new subheading after ”Right to health” This section has been named: ”Effects of conflict on health equity”- see page 7.

* The paragraph “The outcome of interest was --- better paying job [35]” on page 7 should probably be cued earlier, so the reader knows already in the beginning that this is a key focus for the article.

>>> Author Response: Thank you very much for pointing this out. We tend to agree with the reviewer that moving the section referred to in the comment, might help the reader to capture the key focus for the article from the beginning. We have carefully tried to cue this section earlier. Even so, we feel it still fits better where it is right now because the other paragraphs from the introduction focus more on study rationale, literature review, and key concepts - see page 7. To help the reader to capture the key focus for the article from the beginning, we rephrased the following on page 3 - specifically referring to the main study outcome as follows:

”It contributes to the body of knowledge on structural determinants of health inequalities in post-conflict settings which is also the main outcome of this study”.

* I am not convinced by the authors’ arguments to exclude LGBT+ as a vulnerable group. The Progress-Plus acronym includes Gender/Sex. It seems very peculiar to apply this category only to mean women, and not gender identity and sexual orientation.

>>> Author Response: Thank you very much for coming back to this relevant question. In reality, many people continue to face acute violence and discrimination based on their sexual orientation and gender identity (OHCHR, 2011), and this violence can be exacerbated in post-conflict times. The very complex post-conflict environment is characterized by large numbers of vulnerable people to be absorbed into civilian life. Additionally, there is a reduced functioning of coping mechanisms at the individual, community, and national levels and many other demands to be met. For these reasons and many others (such as the institutional destruction to be addressed), we very much agree with the reviewer’s assessment that LGBTQ+ people belong to the vulnerable population sub-groups in post-conflict environments. However, it might be more relevant to place this into the right perspective by defining what is meant by post-conflict vulnerability, who is vulnerable, and to what particular risk? 

How gender identity and sexual orientation (about inequality in health) should be handled in a post-conflict context is indeed an important issue. As noted previously, the focus of our review was to collate the current knowledge and knowledge gaps about structural determinants of health inequalities within post-conflict settings and assess the effects of approaches aimed at addressing structural determinants of health inequalities in this environment. In particular, we found sufficient empirical evidence on gender identity in our review, but less on sexual orientation concerning inequality in health, suggesting that this important issue cannot be easily summarized in the current manuscript. Luckily, we can already learn about this topic from a stable (non-conflict) context. Findings from a recent comprehensive UK scoping review about the Politics of LGBT+ Health Inequality, Elizabeth McDermott et al. (2021) notes that the current body of UK LGBT+ health inequality research is relatively small but there is clear evidence of health inequities between the LGBT+ community, suggesting that the evidence base is insufficient to address this nationally recognized health inequality. Thus, although many people agree (and we are among them) and recognize that LGBTQ+ people as part of the most vulnerable populations (particularly in post-conflict settings) - they require special attention when developing public health interventions that can successfully tackle LGBT+ health inequality. Special consideration must be given to the particular needs and risks faced by individuals based on their sexual orientation and gender identity (e.g. LGBTQ+ people). 

Because the reviewer is a strong proponent of including LGBTQ+ people as a separate sub-group in our review, we would not feel comfortable discussing this sub-group without giving it proper attention in the search terms used to identify publications. Thus, this would simply require running an entirely new search and writing a new paper on this (which we may consider doing in the future given the socially relevant topic). For now, we prefer to point the reader in the direction of the sub-groups we have identified as being most vulnerable in this environment.

* The review should not include the reference to Alzate (2008). The study selection criteria on page 9 say “a) post-conflict environment was explicitly described as study setting”. In the early 2000s, Colombia had the highest levels of conflict-related mortality. On page 17, the authors write: “From a human rights perspective, [42] notes that IDPs have been denied the opportunity to participate in the development process or their engagement has been seriously compromised in Colombia.” Alzate’s article does not discuss IDPs in a post-conflict scenario (which by any standard definitions, would be after the Havana Peace Accords signed in 2016). The Peace Accords have instated many programs to address health needs of IDPs. This article is just too old to be relevant for the case of Colombia, if the authors are indeed selecting on post-conflict settings.

>>> Author Response: We thank the reviewer for this valid comment and the suggestions. Although the paper by Alzate was initially included because it fulfilled the search criteria, we removed it. Furthermore, we still want to emphasize that on page 6 of our manuscript, we referred to additional criteria for the final selection of publications including - (i) containing at least one of the three key concepts (post-war, inequality, and health) (S2 File).

Indeed, the reviewer is right by pointing out that Colombia shouldn’t be seen as a post-conflict setting by the time Alzate’s article was published in 2008. One argument supporting this statement is that we didn’t find a signed peace agreement (part of the characteristics used to define post-conflict settings in our manuscript) this time.

To better understand the history of the conflict in Colombia, we checked and found the classification by Ramos Jaraba et al. (2020) relevant to this discussion - see summary below:

Period/years Description

FIRST PERIOD (1958–1982) Emergence of guerrilla forces, subversive violence, and violence between two political parties.

SECOND PERIOD (1982–1996) Was defined by the military strengthening of guerrilla forces and their territorial expansion, as well as the appearance of paramilitaries and proliferation of drug trafficking. Also there were peace process with some guerrilla groups.

THIRD PERIOD (1996–2005) Exhibited the greatest upsurge in the conflict with the expansion of illegal armed groups, massacres, extortion, kidnapping, drug trafficking.

FOURTH PERIOD (2005–2012) Known as the period of “readjustment of the armed conflict,” was defined by two crucial processes: 1) a military offensive by the Colombian state, and 2) negotiations and subsequent demobilization of paramilitary groups.

FIFTH PERIOD (2012–2017) Peace dialogues in Havana between the national government and FARC-EP guerillas, dialogue with ELN guerillas in Quito, reconfiguration of other armed groups in the country.

Alzate’s paper was published in the fourth period of the long conflict. Although this period does not refer to a signed peace agreement in Colombia, it indicates that negotiations and demobilization of paramilitary groups were ongoing.

Finally, we addressed all comments and removed the reference to Alzate (2008) through the manuscript. We also updated supplemental study materials (PRISMA and extraction table) to reflect this change.

* Reference 95 on page 16 about prenatal exposure is not correct. In this article, Roseboom writes “In fact, epigenetic studies have shown that prenatal exposure to violence leaves lasting epigenetic marks on the glucocorticoid receptor that can still be detected 10–19 years later.” Why do you refer to a literature review, and not the empirical article? Also, the wording of the sentence is a clear case of copy-paste. (The authors write: “Similarly, [95] notes that prenatal exposure to violence leaves lasting epigenetic marks on the glucocorticoid receptor that can still be detected 10–19 years later.”) I strongly encourage the authors to use their own words, and not refer to literature reviews but directly to empirical articles.

>>> Author Response: Thank you very much for your constructive feedback. We have addressed your comments as described below.

Page 19. In a study of the effects of the conflict on mortality both in utero and during the first year of life in the DRC, Dagnelie and colleagues [64] show that in utero exposure to conflict negatively affects the number of males (by a reduction of the number of male fetuses) reaching birth. Furthermore, the authors show that conflict exposure increases infant mortality specifically among girls. Possible explanations for such an outcome are in utero biological factors. 

In another study discussing why achieving gender equality is of fundamental importance to improve the health and well-being of future generations, Roseboom (94) notes that prenatal exposure to violence has been associated with long-term effects on children’s health, especially on brain structure and neural function, that can still be detected many years later.

* On page 21 and many other places in the article, the authors refer to the general population as a population sub-group. A general population cannot also be a population sub-group. Please rephrase.

>>> Author Response: Thank you for constructive feedback. For clarification, this has now been rephrased and changed to “civilian population ”. Accordingly, it has also been modified throughout the revised manuscript.

* The discussion seems quite repetitive/similar to the results. I suggest the authors to merge the discussion section until mid-page 26 into the results, and to let the parts starting with the “Study Strenghts and Limitations” be the discussion.

>>> Author Response: We understood that clarifications were needed to avoid repetitions in the results section. As suggested by the reviewer, we merged the discussion section until mid-page 26 into the results. As a result, the parts starting with the “Study Strengths and Limitations” are now in the discussion section.

* In reference to the authors’ rebuttal with respect to IDPs versus refugees, I want to point out that not all international refugees enjoy protection from the UNHCR.

>>> Author Response: Thanks for your comment. We agree with the reviewer that not all international refugees enjoy protection from the UNHCR.

* On page 12, the authors write “Three regions, including Asia Pacific, South America, and Europe, had comparable data …”. What does “comparable data” mean? Please clarify.

>>> Author Response: Thank you very much for pointing this out. We rephrased the sentence to explain what we mean by comparable data. The revised text on page 13 reads as follows: ”

”Small numbers of papers emanated from three regions, including Asia Pacific, South America, and Europe – potentially an indicator that recent peace agreements have been reached in these regions. Only one study collected data in the Russia and Eurasia region.”

---

## [Decision Letter · Decision Letter 2]

31 Jan 2022

PONE-D-21-08382R2Health inequalities in post-conflict settings: A systematic review.PLOS ONE

Dear Dr. Bwirire,

Thank you for submitting your manuscript to PLOS ONE. After careful consideration, we feel that it has merit but does not fully meet PLOS ONE’s publication criteria as it currently stands. Therefore, we invite you to submit a revised version of the manuscript that addresses the points raised during the review process.

We look forward to receiving your revised manuscript.

Kind regards,

Jakob Pietschnig, PhD, MSc

Academic Editor

PLOS ONE

Journal Requirements:

Reviewers' comments:

Reviewer's Responses to Questions

**Comments to the Author**

1. If the authors have adequately addressed your comments raised in a previous round of review and you feel that this manuscript is now acceptable for publication, you may indicate that here to bypass the “Comments to the Author” section, enter your conflict of interest statement in the “Confidential to Editor” section, and submit your "Accept" recommendation.

Reviewer #1: All comments have been addressed

2. Is the manuscript technically sound, and do the data support the conclusions?

Reviewer #1: Yes

3. Has the statistical analysis been performed appropriately and rigorously? 

Reviewer #1: N/A

4. Have the authors made all data underlying the findings in their manuscript fully available?

Reviewer #1: Yes

5. Is the manuscript presented in an intelligible fashion and written in standard English?

Reviewer #1: Yes

6. Review Comments to the Author

Reviewer #1:

This manuscript has been much improved. For example, I particularly like that the authors use the term “civilian population” rather than “general population” and explain their rationale for this.

Some minor comments remain before the manuscript is ready for publication:

On inclusion/exclusion of LGBT as a vulnerable group: the authors’ counterarguments are fine, but then I suggest adding this as a limitation and a gap for future work. Picking two or three sentences from the rebuttal letter would suffice.

The paragraph on page 4 starting with: ”There is evidence that conflict can be a primary driver of health inequalities (9). The following example from the DRC clearly illustrates this point.” What does this example have to do with health inequalities? It’s more about the difference between direct and indirect mortality in conflict. For it to have something to do with health inequalities, it would have needed information about how mortality differs across SES groups.

Page 13 “… potentially an indicator that recent peace agreements have been reached in these regions.” This can easily be checked, so there is no need for speculation. See for example https://www.peaceagreements.org/. I would assume Sub-Saharan Africa is more researched due to the many international development cooperation initiatives in the region.

Page 17 “According to Sorenson (115), one feature of post-conflict environments that is often ignored by academics and policymakers is the situation of women.” What does this mean? Women’s situation in post-conflict has been extensively researched.

References # 14 and # 105 (and possibly others) are not complete.

Last sentence on page 5: “throughout” – not “thought”.

7. PLOS authors have the option to publish the peer review history of their article (what does this mean?). If published, this will include your full peer review and any attached files.

Reviewer #1: No

---

## [Author Response · Author response to Decision Letter 2]

16 Feb 2022

Rebuttal letter

Response to Reviewer(s)

PONE-D-21-08382R2

Re: Resubmission of manuscript “Health inequalities in post-conflict settings: A systematic review”.

PLOS ONE

We would like to thank you and the reviewers for the opportunity to revise and resubmit our manuscript to PLOS ONE. We further appreciate your input and we have addressed the remaining minor comments. We have copied the review decision from the submission menu of the editorial manager and used this format to write our responses. Below we provide the point-by-point responses to the comments raised. All modifications in the revised manuscript have been highlighted (with Track Changes). Of course, we remain at your disposal should you have any further suggestions for improvements to our paper. 

Dear Dr. Bwirire,

Thank you for submitting your manuscript to PLOS ONE. After careful consideration, we feel that it has merit but does not fully meet PLOS ONE’s publication criteria as it currently stands. Therefore, we invite you to submit a revised version of the manuscript that addresses the points raised during the review process.

We look forward to receiving your revised manuscript.

Kind regards,

Jakob Pietschnig, PhD, MSc

Academic Editor

PLOS ONE

Journal Requirements:

Reviewers' comments:

Reviewer's Responses to Questions

Comments to the Author

1. If the authors have adequately addressed your comments raised in a previous round of review and you feel that this manuscript is now acceptable for publication, you may indicate that here to bypass the “Comments to the Author” section, enter your conflict of interest statement in the “Confidential to Editor” section, and submit your "Accept" recommendation.

Reviewer #1: All comments have been addressed

2. Is the manuscript technically sound, and do the data support the conclusions?

Reviewer #1: Yes

3. Has the statistical analysis been performed appropriately and rigorously? 

Reviewer #1: N/A

4. Have the authors made all data underlying the findings in their manuscript fully available?

Reviewer #1: Yes

5. Is the manuscript presented in an intelligible fashion and written in standard English?

Reviewer #1: Yes

6. Review Comments to the Author

Reviewer #1:

This manuscript has been much improved. For example, I particularly like that the authors use the term “civilian population” rather than “general population” and explain their rationale for this.

Some minor comments remain before the manuscript is ready for publication:

Reviewers' comment:

On inclusion/exclusion of LGBT as a vulnerable group: the authors’ counterarguments are fine, but then I suggest adding this as a limitation and a gap for future work. Picking two or three sentences from the rebuttal letter would suffice.

>>> Author Response:

We thank the reviewer for this valid comment and the suggestions. The revised text on page 27 of our manuscript reads now as follows: 

”Third, although we agree and recognize LGBTQ+ (Lesbian, Gay, Bisexual, Trans, Queer, and others) people as belonging to the most vulnerable sub-groups in post-conflict settings – this sub-group was not included in our systematic review because LGBTQ+ issues require special attention. For the same reason, they could not be easily summarized in the current manuscript. To successfully tackle LGBTQ+ issues, special consideration must be given to the particular needs and risks faced by individuals based on their sexual orientation and gender identity. Future systematic reviews on health inequalities within the post-conflict environment should also include LGBTQ+ people as a vulnerable sub-group”. 

Reviewers’ comment:

The paragraph on page 4 starting with: ”There is evidence that conflict can be a primary driver of health inequalities (9). The following example from the DRC clearly illustrates this point.” What does this example have to do with health inequalities? It’s more about the difference between direct and indirect mortality in conflict. For it to have something to do with health inequalities, it would have needed information about how mortality differs across SES groups.

>>> Author Response:

We thank the reviewer for raising this question. We went back to the paper in which the relionship between conflict and health inequalities has been extensively discussed (Ranson et al, 2007). Specifically, we focused on the broad context of interest for our systematic review, which includes inequality in health within the post-conflict environment. While reading Ranson’s review again - a few arguments came up that explain why we selected the example from the DRC to illustrate the link between conflict and health inequalities (which we have made explicit in the revised version of the manuscript – as described below). 

First, we agree with the reviewer that socio-economic status (as measured by education, income or occupation) is a key determinant of health. However, there are other relevant determinants of health which are specific to the post-conflict environment, such as: 1) differential exposure – which may vary (by type, amount, and duration) between social groups 2) differential vulnerability – even when a given risk factor is distributed evenly across social groups, its impact on health may be unevenly distributed due to underlying differences between social groups in their vulnerability or susceptibility to that factor and 3) differential consequences – the impact of a certain health event differs depending on an individual’s or family’s socio-economic circumstances or health. 

Second, life expectancy and mortality trends are familiar ways of measuring health status and monitoring health inequalities. The example of the DRC illustrates the links between conflict and geographical inequalities in mortality, suggesting that mortality rate was higher in unstable eastern provinces, and supporting the idea that life expectancy is lower in more deprived areas.

To clearly explain this point to the reader, we have revised the paragraph on page 4 of the manuscript, emphasizing the link between conflict and health inequalities in the DRC, and we have rephrased this sentence as follows: 

”There is evidence that conflict can be a primary driver of health inequalities (9). Specifically, socio-economic status (as measured by education, income or occupation) is a key determinant of health. But, there are other more important determinants of health which are specific to the post-conflict environment, such as : 1) differential exposure – which may vary (by type, amount, and duration) between social groups 2) differential vulnerability – even when a given risk factor is distributed evenly across social groups, its impact on health may be unevenly distributed due to underlying differences between social groups in their vulnerability or susceptibility to that factor and 3) differential consequences – the impact of a certain health event differs depending on an individual’s or family’s socio-economic circumstances or health. 

Furthermore, life expectancy and mortality trends are familiar ways of measuring health status and monitoring health inequalities. Countries without basic data on mortality (including post-conflict countries), by socioeconomic indicators might have difficulties progressing their health equity agenda. The example of the DRC illustrates the links between conflict and regional inequalities in mortality. It indicates that mortality rate was higher in unstable eastern provinces compared to stable provinces, and suggest that life expectancy was lower in more deprived areas.

Through a series of mortality surveys conducted between 2000 and 2004, the International Rescue Committee (IRC) has documented the humanitarian impact of war and conflict in the DRC. These studies estimated that 3.9 million people had died since 1998, arguably making the DRC's conflict the world's deadliest crisis since World War II. Less than 10 percent of all deaths were due to violence, with most deaths attributed to easily preventable and treatable health conditions (indirectly caused by conflict- and resulting in inequitable health) such as malaria, diarrhea, respiratory infections, and malnutrition (10). Furthermore, deaths are often used as the primary indicator for the effect that conflict has on civilians; indeed, the number of these deaths declined in the DRC following an increase in security (11-13).”

Reviewers' comment:

Page 13 “… potentially an indicator that recent peace agreements have been reached in these regions.” This can easily be checked, so there is no need for speculation. See for example https://www.peaceagreements.org/. I would assume Sub-Saharan Africa is more researched due to the many international development cooperation initiatives in the region.

>>> Author Response:

Thank you for sharing this very interesting hyperlink about peace agreements status. We checked the Peace Agreement Database (https://www.peaceagreements.org/) and found relevant information about the specific regions of interest (e.g.: Asia Pacific, South America, and Europe). Between 1990 and 2021, 381 peace agreements were signed in the Asia Pacific, 414 in Europe, and 195 in the Americas - which support our previous reporting that recent peace agreements were reached in these regions. For the sake of clarity, we have rephrased this in the revised version of the manuscript on Page 13: “…an indicator that recent peace agreements have been reached in these regions ”, omitting the ‘potentially’.

Reviewers' comment:

Page 17 “According to Sorenson (115), one feature of post-conflict environments that is often ignored by academics and policymakers is the situation of women.” What does this mean? Women’s situation in post-conflict has been extensively researched.

>>> Author Response: 

Thank you very much for your question. We agree with the reviewer that women’s situation in post-conflict has been extensively researched and findings suggest that their situation is characterized by fundamental weaknesses or imperfection in several ways (e.g. Targeting women for professional health care training is a key to improving the community's health status, In addition, it may also provide valuable opportunities for long-term employment. However, in practice, many programs continue to discriminate against women when enrolling people in training activities). This has serious implications for both our understanding of women's situation and our ability to effectively and efficiently assist them. 

To explain what we mean by “often ignored”; we revised the sentence on page 17 as follows:

“According to Sorenson (115), health care and other social facilities remain inadequate for women’s health in the post-conflict period. As such, they face new challenges (e.g. their traditional roles and relationship) and inherit additional responsibilities. The social transformation occurring in the post-conflict context opens up opportunities that should not be missed for women ”. 

Reviewers' comment:

References # 14 and # 105 (and possibly others) are not complete.

>>> Author Response:

Thank you very much for the reminder. For completeness, missing details were added to references # 14 and #105 in EndNote. Furthermore, relevant details were added to references #7, #10, #13, #16, #18, #20, #21, #22, #23, #28, #32, #37, #59 and #105 in the revised manuscript. 

Reviewers' comment:

Last sentence on page 5: “throughout” – not “thought”.

>>> Author Response: 

Thank you for this comment. We have made revisions accordingly. The original sentence on page 5 of the manuscript has been modified with the inclusion of the correct word: ”throughout”. As a result we believe the message now is more clear.

7. PLOS authors have the option to publish the peer review history of their article (what does this mean?). If published, this will include your full peer review and any attached files.

Do you want your identity to be public for this peer review? For information about this choice, including consent withdrawal, please see our Privacy Policy.

Reviewer #1: No

---

## [Editor Report · Decision Letter 3]

23 Feb 2022

Health inequalities in post-conflict settings: A systematic review.

PONE-D-21-08382R3

Dear Dr. Bwirire,

We’re pleased to inform you that your manuscript has been judged scientifically suitable for publication and will be formally accepted for publication once it meets all outstanding technical requirements.

Kind regards,

Jakob Pietschnig, PhD, MSc

Academic Editor

PLOS ONE
---

## [Editor Report · Acceptance letter]

4 Mar 2022

PONE-D-21-08382R3 

Health inequalities in post-conflict settings: A systematic review. 

Dear Dr. Bwirire:

I'm pleased to inform you that your manuscript has been deemed suitable for publication in PLOS ONE. Congratulations! Your manuscript is now with our production department. 

Kind regards, 

on behalf of

Dr. Jakob Pietschnig 

Academic Editor

PLOS ONE